# Ensembling Diffusion Models via Adaptive Feature Aggregation

**Cong Wang**[1*†]    **Kuan Tian**[2†]    **Yonghang Guan**[2]    **Fei Shen**[2]    **Zhiwei Jiang**[1‡]
**Qing Gu**[1]    **Jun Zhang**[2‡]

[1] State Key Laboratory for Novel Software Technology, Nanjing University
[2] Tencent AIPD

cw@smail.nju.edu.cn, {kuantian, yohnguan, ffeishen}@tencent.com,
{jzw, guq}@nju.edu.cn, junejzhang@tencent.com

## Abstract

The success of the text-guided diffusion model has inspired the development and release of numerous powerful diffusion models within the open-source community. These models are typically fine-tuned on various expert datasets, showcasing diverse denoising capabilities. Leveraging multiple high-quality models to produce stronger generation ability is valuable, but has not been extensively studied. Existing methods primarily adopt parameter merging strategies to produce a new static model. However, they overlook the fact that the divergent denoising capabilities of the models may dynamically change across different states, such as when experiencing different prompts, initial noises, denoising steps, and spatial locations. In this paper, we propose a novel ensembling method, Adaptive Feature Aggregation (AFA), which dynamically adjusts the contributions of multiple models at the feature level according to various states (i.e., prompts, initial noises, denoising steps, and spatial locations), thereby keeping the advantages of multiple diffusion models, while suppressing their disadvantages. Specifically, we design a lightweight Spatial-Aware Block-Wise (SABW) feature aggregator that adaptive aggregates the block-wise intermediate features from multiple U-Net denoisers into a unified one. The core idea lies in dynamically producing an individual attention map for each model's features by comprehensively considering various states. It is worth noting that only SABW is trainable with about 50 million parameters, while other models are frozen. Both the quantitative and qualitative experiments demonstrate the effectiveness of our proposed method.[1]

## 1 Introduction

Diffusion models (Sohl-Dickstein et al., 2015; Ho et al., 2020) have progressively become the mainstream models for text-guided image generation (Nichol et al., 2021; Ramesh et al., 2022; Saharia et al., 2022; Rombach et al., 2022; Balaji et al., 2022; Xue et al., 2023; Feng et al., 2023), which treats generation as an iterative denoising task. Recently, the open-source stable diffusion (SD) (Rombach et al., 2022) model has prompted the development and release of numerous powerful diffusion models within the open-source community (e.g., CivitAI[2]). These models are typically fine-tuned on various expert datasets, showcasing diverse denoising capabilities. Leveraging multiple high-quality models to dig out better generations is an important research direction, which has not been extensively studied.

Existing methods leverage multiple diffusion models through the weighted merging of model parameters, which can be called the static method. The weights are usually manually set (e.g., Weighted-

---

*Work done during an internship at Tencent AIPD.
†Equal contribution.
‡Corresponding authors.

[1]The code is available at https://github.com/tenvence/afa.
[2]https://civitai.com/

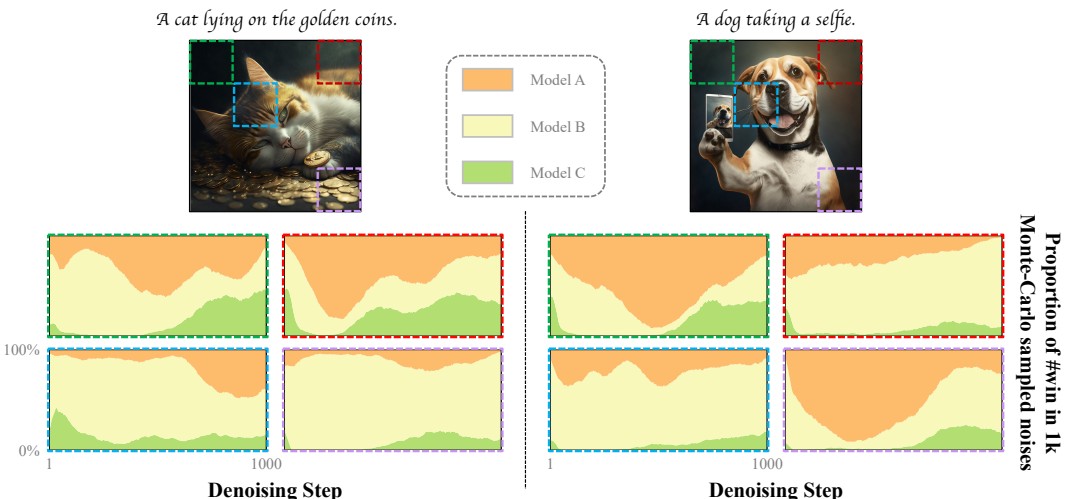

Figure 1: Examples to illustrate the dynamical change of the denoising capabilities across various states. We conduct experiments on different prompts with various initial noises. We then plot the proportion of *wins* (i.e., the model with the least error between the predicted noise and the initial noise), for each model in a certain spatial region.

Merging[3] and MBW[4]) or automatically searched through enumeration (e.g., autoMBW[5]). In contrast, model ensembling, which can be called the dynamic method, often uses dynamic strategies to fuse multiple models at the feature level. Unlike ensembling for classification models (Freund & Schapire, 1995) that usually work after the output, ensembling for diffusion models typically needs to work for each block. However, the denoising capabilities of models vary not only at different blocks but also at different spatial locations. As illustrated in Figure 1, we conduct denoising experiments on different prompts with various initial noises. Then, we plot the proportion of *wins* (i.e., the model with the least error between the predicted noise and the initial noise), for each model in a certain spatial region, to distinguish the denoising capabilities of the models at different states. In other words, different prompts, initial noises, denoising steps, and spatial locations can all have a significant impact on the denoising capabilities of diffusion models. This implies that an adaptive method is needed to ensure that each diffusion model dominates the generation at its strongest states.

In this paper, we propose a novel Adaptive Feature Aggregation (AFA) method, which dynamically adjusts the contributions of multiple models at the feature level by taking into account various states, such as prompts, initial noises, denoising steps, and spatial locations. Specifically, we design a lightweight Spatial-Aware Block-Wise (SABW) feature aggregator that adaptively aggregates the block-level intermediate features from multiple U-Net denoisers into a unified one. The core idea of adaptive aggregation lies in dynamically producing an individual attention map for each model's features by comprehensively taking into account the various states. A noteworthy aspect of AFA is that only SABW is trainable with about 50 million parameters, while all other models are frozen.

Our main contributions are summarized as follows:

- We propose an ensembling-based AFA method to dynamically adjust the contributions of multiple models at the feature level.

- We design the SABW feature aggregator that can produce attention maps according to various states to adaptively aggregate the block-level intermediate features from multiple U-Net denoisers.

- We conduct both quantitative and qualitative experiments, demonstrating that our AFA outperforms the base models and the baseline methods in both superior quality and context alignment.

---

[3] https://github.com/hako-mikan/sd-webui-supermerger
[4] https://github.com/bbc-mc/sdweb-merge-block-weighted-gui
[5] https://github.com/Xerxemi/sdweb-auto-MBW

## 2 RELATED WORK

**Text-Guided Image Synthesis.** Early works for text-guided image synthesis leverage Generative Adversarial Networks (GAN) (Goodfellow et al., 2014) conditioned on text (Tao et al., 2022; Xu et al., 2018; Zhang et al., 2021a; Zhu et al., 2019). Based on the success of Transformers (Vaswani et al., 2017), many subsequent works reframe text-guided image synthesis as a sequence-to-sequence task (Zhang et al., 2021b; Ramesh et al., 2021; Ding et al., 2021; Gafni et al., 2022; Yu et al., 2022). Recently, diffusion models (Sohl-Dickstein et al., 2015; Ho et al., 2020) gradually become mainstream, which treats image generation as an iterative denoising task. By injecting text as a condition into the denoising process, many diffusion-based models achieve significant success in text-guided image synthesis (Nichol et al., 2021; Ramesh et al., 2022; Saharia et al., 2022; Rombach et al., 2022; Balaji et al., 2022; Xue et al., 2023; Feng et al., 2023; Shen et al., 2023a; 2024a;b; Shen & Tang, 2024; Wang et al., 2024; Fu et al., 2024; Shen et al., 2025). Among them, the Latent Diffusion Model (LDM) (Rombach et al., 2022) performs the diffusion and reverse process in the latent space, instead of the pixel space, which largely reduces the computational burden. Among all the implementations of the Latent Diffusion Model, the stable diffusion model (SD) is the most famous one. Along with its public availability, the open-source community achieves tremendous success, with the emergence of many excellent fine-tuned models. These high-quality models are typically fine-tuned on various expert datasets, showcasing diverse denoising capabilities. In this paper, we aim to leverage multiple such models to achieve stronger text-to-image generation.

**Model Merging.** For vision or language understanding tasks, the effectiveness of parameter merging (Frankle et al., 2020; Wortsman et al., 2022b; Matena & Raffel, 2022; Ilharco et al., 2022b; Li et al., 2022; Don-Yehiya et al., 2022; Jin et al., 2022) can be interpreted from the perspectives of loss landscape (Wortsman et al., 2022a; Ainsworth et al., 2022; Stoica et al., 2023) and task arithmetic (Ilharco et al., 2022a; Ortiz-Jimenez et al., 2023; Zhang et al., 2023). However, merging-based methods have not been widely studied for generation tasks, especially diffusion-based generation. Recently, many intuitive merging-based methods have emerged. One of the popular methods is Weighted-Merging, which manually determines weights to merge each U-Net (Ronneberger et al., 2015) parameter of multiple SD models. Although the simplicity, Weighted Merging coarsely allocates weight to all U-Net blocks. As an improvement, Merge Block Weighted (MBW) allows for manually setting different merging weights for the parameters of distinct U-Net blocks, which provides a more fine-grained merging strategy. Furthermore, to reduce the reliance on manual weights, autoMBW attempts to automate the merging process, which enumeratively selects the optimal combination of MBW weights by an aesthetic classifier. However, autoMBW is constrained by the performance bottleneck of the aesthetic classifier, and the enumerative selection leads to a huge time consumption to find the optimal settings. Such merging methods can be called the static methods. In this paper, we aim to leverage multiple diffusion models by the dynamic methods, i.e., ensembling.

**Model Ensembling.** Model ensembling is an effective method to achieve better performance (Zhou, 2012), which has been widely applied in various vision understanding tasks, such as classification (Zhao et al., 2005; Rokach, 2010; Yang et al., 2010), regression (Mendes-Moreira et al., 2012; Ren et al., 2015), clustering (Vega-Pons & Ruiz-Shulcloper, 2011). While fewer works focus on the ensembling of generative models, because of the complexity of image space. Vision-Aided GAN (Kumari et al., 2022) focuses on GANs (Goodfellow et al., 2014), which guides the optimization of a target generator by ensembling pretrained models as a loss. MagicFusion (Zhao et al., 2023) focuses on the diffusion models, which fuses the predicted noises of two expert U-Net denoisers to implement specific applications, such as style transferring and object binding. In this paper, we aim to efficiently ensemble multiple diffusion models to achieve general generation improvements[6].

## 3 METHOD

### 3.1 PRELIMINARY

As a type of generative model, the diffusion model (Sohl-Dickstein et al., 2015; Ho et al., 2020) consists of two processes, which are the diffusion process and the reverse process, respectively. In

---

[6]Note that combining the styles/concepts of multiple diffusion models is also a common goal of ensembling/merging, which is not the interest of our work.

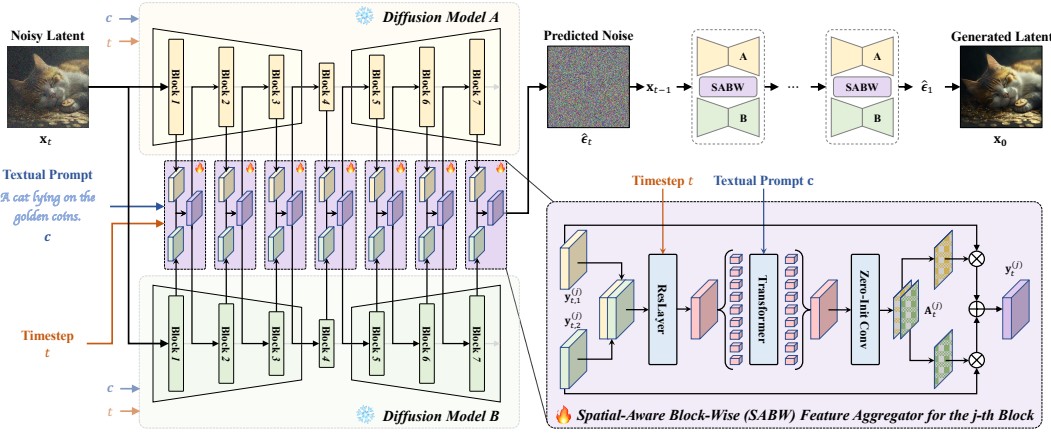

Figure 2: Framework of ensembling multiple diffusion models by our AFA method.

the diffusion process (i.e., the forward process), the Gaussian noises are iteratively added to degrade the images over $T$ steps until the images become completely random noise. In the reverse process, a trained denoiser is used to iteratively generate images from the sampled Gaussian noise.

When training, given an input image $\mathbf{x}_0$ and the additional condition $\mathbf{c}$ (e.g., encoded textual prompts etc.), the denoising loss is defined as

$$\mathcal{L}_{\text{denoise}} = \mathbb{E}_{\mathbf{x}_0, \boldsymbol{\epsilon} \sim \mathcal{N}(\mathbf{0}, \mathbf{I}), \mathbf{c}, t} \left\| \boldsymbol{\epsilon}_\theta \left( \mathbf{x}_t, \mathbf{c}, t \right) - \boldsymbol{\epsilon} \right\|^2 . \tag{1}$$

Among them, $\mathbf{x}_t = \sqrt{\alpha_t} \mathbf{x}_0 + \sqrt{1 - \alpha_t} \boldsymbol{\epsilon}$ is the noisy image at timestep $t \in [1, T]$, where $\alpha_t$ is a predefined scalar from the noise scheduler. $\boldsymbol{\epsilon}$ is the added noise. $\boldsymbol{\epsilon}_\theta$ is the denoiser with the learnable parameters $\theta$, which predicts the noise to be removed from the noisy image.

LDM stands out as one of the most popular diffusion models. It performs the diffusion and reverse process in the latent space which is encoded by a Variational Auto-Encoder (VAE) (Kingma & Welling, 2013; Rezende et al., 2014). In LDM, the U-Net structure (Ronneberger et al., 2015) is used for denoiser, which contains three parts of blocks, which are the down-sampling blocks, the middle block, and the up-sampling blocks, respectively. Each down-sampling block is skip-connected with a corresponding up-sampling block. These blocks are composed of Transformer (Vaswani et al., 2017) layers and ResNet (He et al., 2016) layers. In the Transformer layers, the cross-attention mechanism is employed to incorporate conditional textual prompts, which plays a crucial role in text-guided image generation.

### 3.2 ADAPTIVE FEATURE AGGREGATION

Our proposed Adaptive Feature Aggregation (AFA) ensembles multiple diffusion models that share the same architecture but different parameters. Specifically, SABW is designed to integrate intermediate features from multiple U-Net denoisers at the block level. Figure 2 demonstrates an example of ensembling two diffusion models, and each U-Net denoiser contains seven blocks by SABW. The forward process of AFA can be seen in Appendix A.

Given $N$ diffusion models to be ensembled, whose U-Net denoisers contain $K$ blocks, let $\boldsymbol{\epsilon}_{\theta_i}$ be the $i$-th U-Net denoiser, and $\boldsymbol{\epsilon}_{\theta_i}^{(j)}$ be its $j$-th block. For the timestep $t$, it transforms the input features $\mathbf{x}_t^{(j)}$ into the output feature by

$$\mathbf{y}_{t,i}^{(j)} = \boldsymbol{\epsilon}_{\theta_i}^{(j)} (\mathbf{x}_t^{(j)}, \mathbf{c}, t) . \tag{2}$$

Note that because of the same architecture of all $\boldsymbol{\epsilon}_{\theta_i}$, $\mathbf{y}_{t,i}^{(j)}$ has the same shape, i.e., $\mathbf{y}_{t,i}^{(j)} \in \mathbb{R}^{h_j \times w_j \times c_j}$, where $h_j$, $w_j$, and $c_j$ are the height, width, and channels, respectively. To achieve the output feature for the next block, a block $f_\varphi^{(j)}$ of SABW is employed to aggregate $\{\mathbf{y}_{t,i}^{(j)}\}_{i=1}^N$,

$$\mathbf{y}_t^{(j)} = f_\varphi^{(j)}(\{\mathbf{y}_{t,i}^{(j)}\}_{i=1}^N, \mathbf{c}, t) \in \mathbb{R}^{h_j \times w_j \times c_j} , \tag{3}$$

where $\varphi$ is the parameters of the SABW block. As a whole, the $j$-th ensembled block of these U-Net denoisers can be formulated as

$$\mathbf{y}_t^{(j)} = \mathcal{F}_\Theta^{(j)}(\mathbf{x}_t^{(j)}, \mathbf{c}, t) , \tag{4}$$

where $\Theta$ is the parameter collection of $\varphi$ and all the $\theta_i$. It can be deduced that an ensembled block operates with the same functionality as a single block.

## 3.3 SPATIAL-AWARE BLOCK-WISE FEATURE AGGREGATOR

The Spatial-Aware Block-Wise (SABW) feature aggregator is learned to aggregate output features of each block from multiple U-Net denoisers. We design SABW to learn spatial attention to reweight the contributions of these U-Net denoisers at each spatial position. Specifically, the spatial attention map $\mathbf{A}_t^{(j)}$ for $N$ U-Net denoisers is achieved by

$$\mathbf{F}_t^{(j)} = g_\varphi^{(j)}(\{\mathbf{y}_{t,i}^{(j)}\}_{i=1}^N, \mathbf{c}, t) \in \mathbb{R}^{h_j \times w_j \times N} , \tag{5}$$

$$\mathbf{A}_t^{(j)} = \mathrm{softmax}(\mathbf{F}_t^{(j)}) \in [0, 1]^{h_j \times w_j \times N} , \tag{6}$$

where, $\mathbf{F}_t^{(j)}$ is the output of the learnable network of SABW $g_\varphi$. The ensembled output feature is

$$\mathbf{y}_t^{(j)} = \mathcal{F}_\Theta^{(j)}(\mathbf{x}_t^{(j)}, \mathbf{c}, t) = \sum_{i=1}^N \mathbf{A}_{t,i}^{(j)} \otimes \mathbf{y}_{t,i}^{(j)} , \tag{7}$$

where $\otimes$ is the element-wise multiplication. $\mathbf{A}_{t,i}^{(j)} \in \mathbb{R}^{h_j \times w_j \times 1}$ is the spatial attention map for the $i$-th U-Net denoiser.

Note that the spatial-attention mechanism of SABW differs to the typical multi-head self-attention mechanism (Vaswani et al., 2017), which simply uses three linear layers to project the input features into three separate spaces (i.e., query, key, and value), and then computes the attention map across all the projected features. Our spatial-attention mechanism is based on an important experimental insight as shown in Figure 1. The divergent denoising capabilities of multiple diffusion models are influenced by the prompts and the denoising steps, and they dynamically vary across different spatial locations. Thus, the projection method needs to account for the input prompt and the current denoising step. And it is sufficient to compute the attention map on each spatial location individually. As a result, better generation performance can be achieved by enabling multiple models to collaborate at different spatial locations under the conditions of the input prompt and the current denoising step.

Specifically, based on the denoising loss (i.e., Eq. 1), the denoising capability[7] of $\epsilon_\theta$ for $(\mathbf{x}_0, \mathbf{c})$ at timestep $t$ can be defined as $-\mathbb{E}_{\epsilon \sim \mathcal{N}(\mathbf{0}, \mathbf{I})} \|\hat{\epsilon}_t - \epsilon\|^2$, where $\hat{\epsilon}_t = \epsilon_\theta(\mathbf{x}_t, \mathbf{c}, t)$. For the spatial location $(x, y)$, the denoising capability is formulated as $-\mathbb{E}_{\epsilon \sim \mathcal{N}(\mathbf{0}, \mathbf{I})} \|\hat{\epsilon}_{(x,y)} - \epsilon_{(x,y)}\|^2$, which considers the noise prediction errors at the spatial location $(x, y)$.

For example, as shown in Figure 1, we employ three SD models to present their positional denoising capability on the specific image and textual prompt by Monte-Carlo sampled noises. We demonstrate the results of four patches which indicate that the divergent denoising capabilities of the three models vary at different spatial locations. SABW is learned to incorporate these denoisers to achieve stronger denoising capability at all spatial locations at the block level.

To implement $g_\varphi$ in Eq. 5, SABW applies a ResNet (He et al., 2016) layer to introduce timesteps by adding the time embedding into the feature map, and a Transformer (Vaswani et al., 2017) layer to introduce textual prompts by the cross-attention,

$$\mathbf{Y}_t^{(j)} \leftarrow \mathrm{concat}(\{\mathbf{y}_{t,i}^{(j)}\}_{i=1}^N) \in \mathbb{R}^{h_j \times w_j \times (N \cdot c_j)} , \tag{8}$$

$$\mathbf{H}_t^{(j)} \leftarrow \mathrm{ResLayer}(\mathbf{Y}_t^{(j)} + \gamma(t)) \in \mathbb{R}^{h_j \times w_j \times d} , \tag{9}$$

$$\mathbf{O}_t^{(j)} \leftarrow \mathrm{TransformerLayer}(\mathbf{H}_t^{(j)}, \mathbf{c}) \in \mathbb{R}^{h_j \times w_j \times d} , \tag{10}$$

$$\mathbf{F}_t^{(j)} \leftarrow \mathrm{ZeroInitConv}(\mathbf{O}_t^{(j)}) \in \mathbb{R}^{h_j \times w_j \times N} , \tag{11}$$

---

[7]Compared with the denoising loss in Eq. 1, the denoising capability is defined by adding the negative sign, because of a negative correlation between the denoising capability and the denoising loss.

| | COCO 2017 | | | Draw Bench Prompts | | | |
|---|---|---|---|---|---|---|---|
| | FID ↓ | IS | CLIP-I | CLIP-T | AES | PS | HPSv2 | IR |
| Base Model A | 13.01 | 5.65 | .6724 | .2609 | 5.4102 | 21.6279 | 27.8007 | .3544 |
| Base Model B | 13.45 | 5.43 | .6775 | .2652 | 5.5013 | 21.4624 | 27.7246 | .2835 |
| Base Model C | 12.32 | 6.32 | .6890 | .2566 | 5.4881 | 21.8031 | 27.9652 | .3922 |
| Wtd. Merging | 10.65 | 6.93 | .6861 | .2626 | 5.4815 | 21.7272 | 27.9086 | .3909 |
| MBW | 11.03 | 6.51 | .6870 | .2624 | 5.4812 | 21.7201 | 27.9080 | .3922 |
| autoMBW | 13.35 | 5.51 | .6772 | .2577 | 5.5056 | 21.4785 | 27.8192 | .3672 |
| MagicFusion | 10.53 | 6.85 | .6751 | .2620 | 5.3431 | 21.3840 | 27.8105 | .3317 |
| **AFA (Ours)** | **9.76** | **7.14** | **.6926** | **.2675** | **5.5201** | **21.8263** | **27.9734** | **.4388** |

Table 1: Quantitative comparison for the base models in Group I (i.e., ER, MMR, and RV).

| | COCO 2017 | | | Draw Bench Prompts | | | |
|---|---|---|---|---|---|---|---|
| | FID ↓ | IS | CLIP-I | CLIP-T | AES | PS | HPSv2 | IR |
| Base Model A | 12.12 | 5.66 | .6849 | .2623 | 5.5641 | 21.8013 | 28.0183 | .4238 |
| Base Model B | 12.41 | 5.59 | .6818 | .2580 | 5.5027 | 21.7249 | 28.0343 | .4202 |
| Base Model C | 12.05 | 5.95 | .6638 | .2642 | 5.5712 | 21.4936 | 27.8089 | .3367 |
| Wtd. Merging | 11.53 | 6.56 | .6824 | .2631 | 5.5756 | 21.7516 | 28.0014 | .4387 |
| MBW | 12.06 | 6.42 | .6826 | .2632 | 5.5772 | 21.7487 | 28.0029 | .4396 |
| autoMBW | 12.39 | 5.62 | .6774 | .2588 | 5.5478 | 21.5135 | 27.9873 | .3513 |
| MagicFusion | 11.63 | 7.13 | .6790 | .2640 | 5.4674 | 21.4270 | 27.9608 | .4194 |
| **AFA (Ours)** | **10.27** | **7.42** | **.6855** | **.2717** | **5.5798** | **21.8059** | **28.0371** | **.4892** |

Table 2: Quantitative comparison for the base models in Group II (i.e., AR, CR, and RCR).

where $\gamma$ is used to map the timestep into the time embedding with the same shape of $\mathbf{Y}_t^{(j)}$. $d$ is the hidden state dimension. ZeroInitConv($\cdot$) is the convolution layer initialized by zero, which leads to equal attention weights for all denoisers before training. We aspire for the model to commence training from a completely equitable state.

## 3.4 TRAINING

Given the ensembled denoiser $\mathcal{F}_\Theta$, the training object is same as the denoising loss in Eq. 1,

$$\mathcal{L} = \mathbb{E}_{\mathbf{x}_0, \boldsymbol{\epsilon} \sim \mathcal{N}(\mathbf{0},\mathbf{I}), \mathbf{c}, t} \left\| \mathcal{F}_\Theta \left( \mathbf{x}_t, \mathbf{c}, t \right) - \boldsymbol{\epsilon} \right\|^2 . \tag{12}$$

Note that the parameters $\Theta$ contains both the parameters of SABW $\varphi$ and the parameters of all U-Net denoisers $\{\theta_i\}_{i=1}^N$, where $\varphi$ are learnable, and $\{\theta_i\}_{i=1}^N$ are frozen.

A well-trained AFA ensembles multiple U-Net denoisers to enhance the denoising capability in every block when experiencing different prompts, initial noises, denoising steps, and spatial locations. It leads to smaller denoising errors and stronger generation performance.

## 3.5 INFERENCE

The inference of AFA starts from the sampled Gaussian noise. Then, the diffusion scheduler (e.g., DDIM (Song et al., 2020), PNDM (Liu et al., 2022), DPM-Solver (Lu et al., 2022a;b), etc.) is applied to generate images with multiple denoising steps.

For each inference step, the noise prediction relies on the technique of Classifier-Free Guidance (CFG) (Ho & Salimans, 2022), which is formulated as $\hat{\boldsymbol{\epsilon}}_t^{\text{pred}} = \hat{\boldsymbol{\epsilon}}_t^{\text{uc}} + \beta_{\text{CFG}}(\hat{\boldsymbol{\epsilon}}_t^{\text{c}} - \hat{\boldsymbol{\epsilon}}_t^{\text{uc}})$ . Among them, $\hat{\boldsymbol{\epsilon}}_t^{\text{pred}}$, $\hat{\boldsymbol{\epsilon}}_t^{\text{uc}}$, $\hat{\boldsymbol{\epsilon}}_t^{\text{c}}$, and $\beta_{\text{CFG}}$ are the predicted noise, the predicted noise with condition, the predicted noise without condition, and a guidance scale, respectively. The latent for the next step is $\mathbf{x}_{t-1} = \frac{1}{\sqrt{\alpha_t}} \mathbf{x}_t - \frac{\sqrt{1-\alpha_t}}{\sqrt{\alpha_t}} \hat{\boldsymbol{\epsilon}}_t^{\text{pred}}$ . Finally, the generated image is achieved from the latent by a VAE decoder.

## 4 EXPERIMENTS

### 4.1 EXPERIMENT SETTINGS

**Base Models.** We select six popular models with the same architecture from SDv1.5[8] in CivitAI. There are 12 down-sampling blocks, 1 middle block, and 12 up-sampling blocks for each model. These models are randomly divided into two groups, with three models in each group. The models from the same group will be ensembled.

- **Group I** contains EpicRealism[9] (ER), MagicMixRealistic[10] (MMR), and RealisticVision[11] (RV);

---

[8] https://huggingface.co/stable-diffusion-v1-5/stable-diffusion-v1-5
[9] https://civitai.com/models/25694?modelVersionId=134065
[10] https://civitai.com/models/43331?modelVersionId=176425
[11] https://civitai.com/models/4201?modelVersionId=130072

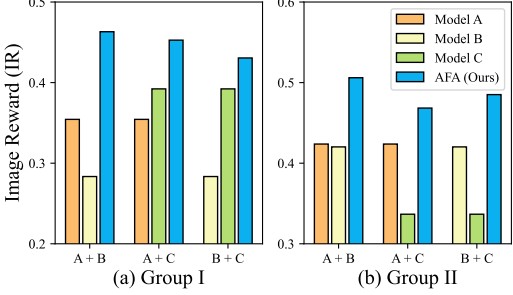 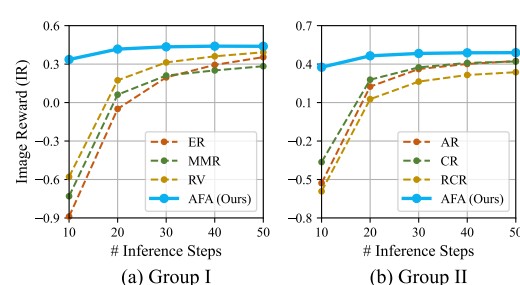

Figure 3: Quantitative comparison between AFA with the two base models.

Figure 4: Effect of varying inference steps.

- **Group II** contains AbsoluteReality[12] (AR), CyberRealistic[13] (CR), and RealCartoonReslistic[14] (RCR).

**Training.** AdamW (Loshchilov & Hutter, 2017) is used as the optimizer with a learning rate of 0.0001 and a weight decay of 0.01. Note that only the parameters of SABW are optimized, while those of U-Net denoisers are frozen. Our AFA framework is trained on 10,000 samples from the dataset JourneyDB (Pan et al., 2023) for 10 epochs with batch size 8. To enable CFG, we use a probability of 0.1 to drop textual prompts.

**Evaluation Protocols.** For a fair comparison, all the methods generate 4 images by DDIM Ho et al. (2020) for 50 inference steps. The CFG weight $\beta_{CFG}$ is set to 7.5. We evaluate the models with two datasets, which are COCO 2017 (Lin et al., 2014) and Draw Bench Prompts (Saharia et al., 2022), respectively. For COCO 2017, we apply four metrics, which are Fréchet Inception Distance (FID), Inception Score (IS), CLIP-I, and CLIP-T, respectively. For Draw Bench Prompts, we apply 4 evaluation metrics, which are AES[15], Pick Score (PS) (Kirstain et al., 2023), HPSv2 (Wu et al., 2023), and Image Reward (IR) (Xu et al., 2023), respectively. More details can be found in Appendix C.

**Baselines.** We compare our AFA with several methods, including three merging-based methods (i.e., Weighted-Merging, MBW, and autoMBW) and one ensembling-based method (i.e., MagicFusion (Zhao et al., 2023)). The evaluation of these baselines follows the protocols.

## 4.2 QUANTITATIVE COMPARISON

The results of the two evaluation protocols are summarized in Tables 1 and 2. Generally, compared with base models, performance improvements are achieved by our AFA. For COCO 2017, after being ensembled by AFA, FID, IS, CLIP-I, and CLIP-T exhibit enhancements under both Group I and Group II. This indicates that the quality of our generated images is more closely with the dataset images. Additionally, the contexts of our generated images align more closely with those of the dataset images and corresponding captions. For Draw Bench Prompts, all four metrics show improvements under both Group I and Group II. The improvement on AES suggests that the aesthetics of our generated images surpass those generated by the base models. The improvements on PS, HPSv2, and IR suggest that the general generation capability can be boosted by applying AFA. Overall, the quantitative results validate the effectiveness of our AFA in ensembling diffusion models.

While MagicFusion attains impressive performance in style-transferring and object-binding by ensembling two U-Net denoisers, it fails to achieve superior general generation when ensembling multiple U-Net denoisers. The merging-based methods, such as Weighted Merging, MBW, and autoMBW, do not always produce models with superior generation capabilities. For instance, under Group II, Weighted Merging achieves a better IR compared to the base models, but under Group I, it yields a lower IR. This could be attributed to the fixed contributions of merged models, which are determined by a set of predefined merging weights. Consequently, these methods fail to effectively

---

[12] https://civitai.com/models/81458?modelVersionId=108576
[13] https://civitai.com/models/15003?modelVersionId=89680
[14] https://civitai.com/models/97744?modelVersionId=104496
[15] https://github.com/christophschuhmann/improved-aesthetic-predictor

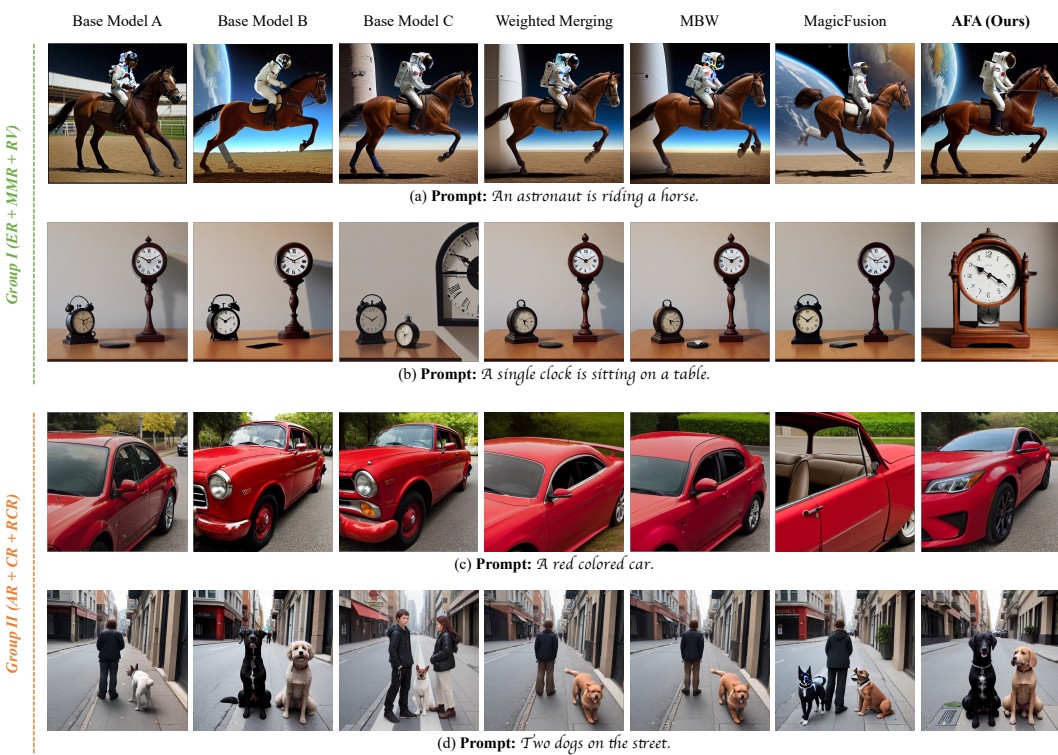

Figure 5: Qualitative comparison between AFA with the base models and the baselines.

facilitate collaboration between models to compensate for each other's weaknesses and improve their generative capabilities. Compared with these baselines, our AFA consistently achieves better performance, demonstrating its superiority.

As depicted in Figure 3, we use AFA to ensemble two models from Group I or Group II, and present the IR scores of base models and the ensembled models on Draw Bench Prompts. The results show that AFA significantly improves the performance when ensembling two U-Net denoisers. The results presented in Table 1, Table 2, and Figure 3 indicate that AFA can be effectively applied to various model combinations and can be extended to accommodate different numbers of models. Meanwhile, it consistently achieves superior generation performance compared to a single base model.

### 4.3 QUALITATIVE COMPARISON

As shown in Figure 5, we present a qualitative comparison of the results from the base models, some of the baseline methods, and our AFA, which allows us to make several insightful observations. Firstly, AFA can generate images with better aesthetics. For example, in Figure 5 (a) and (c), AFA can generate images with improved composition and finer details, compared to both the base models and the baseline methods. Secondly, AFA excels in achieving superior context alignment. For example, in Figure 5 (b), all the base models and the baseline methods generate images containing more than *one clock*, despite the textual prompt specifying *a single clock*. In contrast, only our AFA generates the image with just one clock, accurately reflecting the provided context. Thirdly, AFA can focus on the correct context of the base models and drop out the incorrect context. For example, in Figure 5 (d), only the base model B generates the image with *two dogs*, while the baseline methods do not fully trust the base model B, and generate the images with *people*. Only our AFA fully trusts the base model B, and generates the image with *two dogs*, which aligns with the textual context.

### 4.4 MODEL STUDY

**Visualization of Attention Maps.** As shown in Figure 6, we select to visualize the attention maps of the first and the penultimate (i.e., 24-th) blocks, both of which are learned by ensembling models

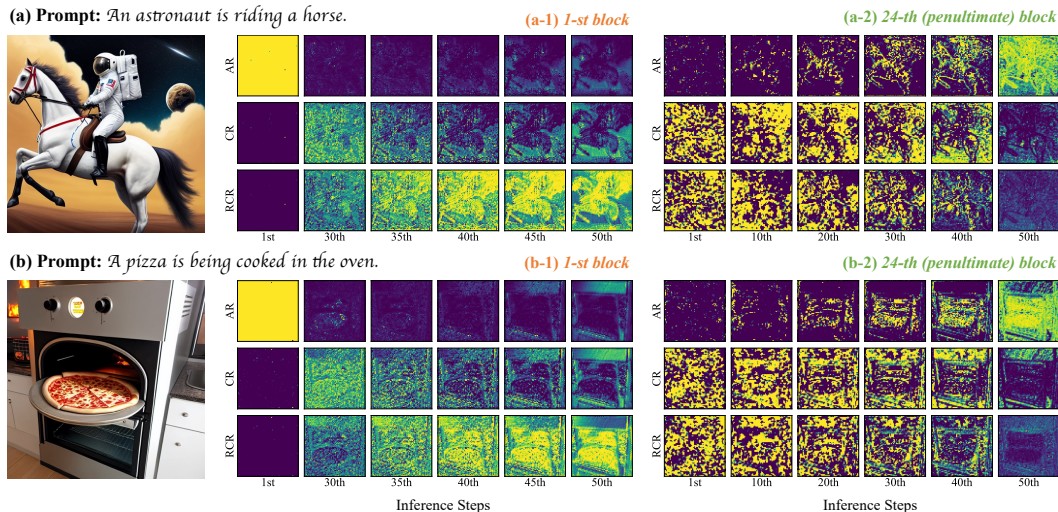

Figure 6: Visualization of the learned attention maps. In each attention map, lighter-coloured positions mean higher attention weights, while darker-coloured positions represent lower weights.

in Group II through our AFA. For the first block, the learned attention maps seem to concentrate solely on the feature map of AR at the first inference stage. However, at the last few inference steps (i.e., from the 30th to the 50th step), the roles of the feature maps from different models begin to diverge. The attention maps tend to highlight the contextual body (i.e., the *astronaut* and the *horse* in Figure 6 (a), and the *pizza* and the *oven* in Figure 6 (b)) in the feature map of RCR, while focusing on the background elements in the feature maps of AR and CR. For the penultimate block, as the inference progresses, the learned attention maps gradually emphasize the contextual body within the feature map of AR, while concentrating on the background within the feature maps of both CR and RCR. Overall, the visualization of the learned attention maps indicates that our AFA can effectively ensemble diffusion models based on the context and the timesteps.

**Effect of Fewer Inference Steps.** As shown in Figure 4 and Figure 7, we investigate the resilience of our AFA to a reduction in the inference steps. Figure 4 clearly shows a significant drop in the performance of the base models when the number of inference steps is reduced. However, the performance of our AFA remains almost stable when the number of inference steps is reduced from 50 to 20, and only experiences a slight decline when the number of inference steps is reduced to 10. Additionally, in Figure 7, we can find that the quality of the images generated by the base models deteriorates with the reduction of inference steps. In contrast, our AFA stably generates high-quality images. Overall, it indicates that greater tolerance for fewer inference steps will be achieved after ensembling by AFA. Furthermore, while a single inference step may require a proportional increase in time, the total time consumed by the entire inference process does not escalate in the same manner.

**Ablation Study.** As shown in Table 3, we conduct the ablation study to investigate the necessity of each component within our AFA. Firstly, we conduct experiments with the following settings. (1) *Only Ensembling Last Block*: The AFA ensembles only the last block, leaving all other blocks unaltered. (2) *Block-Wise Averaging*: Each block is ensembled by averaging the output feature maps, rather than using SABW. (3) *Noise Averaging*: Only the last block is ensembled by averaging the output feature maps (i.e., predicted noises). The observed performance declines in these settings suggest that the design of the AFA effectively contributes to the ensembling of diffusion models, thereby leading to improved generation. Next, we conduct an ablation study on

|  | IR (Group I) | IR (Group II) |
|---|---|---|
| **AFA (Full Model)** | **.4388** | **.4892** |
| Only Ensembling Last Block | .4176 | .4374 |
| Block-Wise Averaging | .4001 | .4372 |
| Noise Averaging | .3919 | .4355 |
| w/o Spatial Location | .4044 | .4610 |
| w/o Timestep | .4235 | .4622 |
| w/o Textual Condition | .4163 | .4559 |

Table 3: Ablation study of AFA.

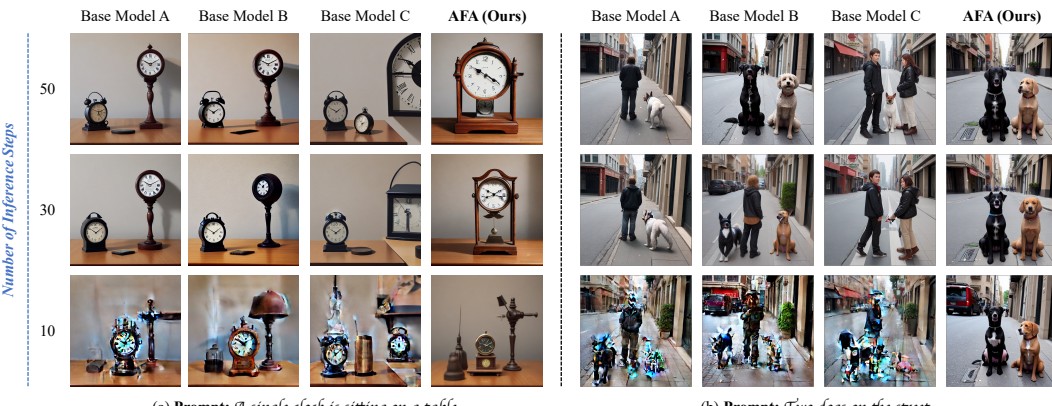

(a) **Prompt:** *A single clock is sitting on a table.*      (b) **Prompt:** *Two dogs on the street.*

Figure 7: Qualitative comparison under different inference steps.

the spatial location from AFA, which learn only a single attention score (i.e., a scalar) for the feature of each U-Net denoiser, as opposed to an attention map. The observed decrease in performance suggests that learning a spatial attention map to adaptively adjust the weights of various U-Net denoisers in the level of spatial location is beneficial for the ensemble of different diffusion models. Finally, we individually ablate the introduced timestep and the textual condition. The decreasing performance indicates that both the timestep and textual condition play crucial roles for AFA.

**Discussion on Computational Efficiency.** (1) *Ensembling Methods.* Compared with another ensembling method (i.e., MagicFusion (Zhao et al., 2023)), the computational efficiency of our AFA for a single inference step aligns closely with it. Both AFA and MagicFusion necessitate running all base models in one inference step. Although SABW in AFA introduces additional parameters, which leads to more computations, it has fewer than the base models (about 1/16). To be precise, our AFA may be marginally less efficient than MagicFusion on single inference steps, but the difference is practically negligible. (2) *Merging Methods.* Because the various models are collapsed into one model, the computational efficiency of the merging methods equals that of a single base model. For the base models and the merging methods, a single image generation takes about 7 seconds with 50 inference steps. When ensembling three such base models, a single generation will take about 22 seconds with the same inference steps. Compared with ensembling, the computation efficiency of one inference step of the merging methods is significantly higher. However, as we discussed in *Effect of Fewer Inference Steps* of Section 4.4, AFA shows remarkable tolerance for fewer inference steps. When limited to 20 inference steps, the performance of AFA does not significantly decline (as shown in Figure 4), while the time consumed is reduced to about 8.8 seconds. This suggests that AFA has comparable computational efficiency with that of the base models or the merging methods throughout the entire inference process.

## 5   CONCLUSION

In this paper, we aim to ensemble multiple diffusion models to achieve better generation performance. To this end, we propose the Adaptive Feature Aggregation (AFA) method, which dynamically adjusts the contributions of multiple models at the feature level according to different prompts, initial noises, denoising steps, and spatial locations. Specifically, we design a lightweight Spatial-Aware Block-Wise feature aggregator that produces attention maps to adaptively aggregate the block-level intermediate features from multiple U-Net denoisers. The quantitative and qualitative experiments demonstrate that AFA achieves improvements in image quality and context alignment.

**Limitation.** A significant limitation of our AFA is that a single inference step may necessitate a proportional increase in inference time, resulting in lower computational efficiency for that step. Fortunately, due to its high tolerance for fewer inference steps, the total time consumed by the entire inference process does not escalate in the same manner. This results in a computational efficiency that is on par with that of the base models or the merging methods.

## 6 ACKNOWLEDGMENTS

This work is supported by the National Natural Science Foundation of China under Grants Nos. 62441225, 61972192, 62172208, 61906085. This work is partially supported by Collaborative Innovation Center of Novel Software Technology and Industrialization. This work is supported by the Fundamental Research Funds for the Central Universities under Grant No.14380001.

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

## A  ALGORITHM OF AFA

The forward process of AFA can be seen in Alg. 1.

---

**Algorithm 1** One forward process of Adaptive Feature Aggregation (AFA).

---

1: **Input:** Timestep $t$, Condition $\mathbf{c}$, Input latent $\mathbf{x}_t$, U-Net denoisers $\{\epsilon_{\theta_i}\}_{i=1}^N$, each of which contains $K$ blocks, and SABW $f_\varphi^{(j)}$ for $j$-th U-Net block.
2: **Output:** Predicted noise $\hat{\epsilon}_t$.
3: Initialize an empty stack $\mathcal{S}$ to store the features from the skip connections;
4: $\mathbf{x}_t^{(1)} \leftarrow \mathbf{x}_t$;
5: **for** block index $j$ from 1 to $K$ **do**
6:     ▷ *Ensembling the $j$-th U-Net block by SABW feature aggregator $f_\varphi^{(j)}$*
7:     **for** U-Net denoiser index $i$ from 1 to $N$ **do**
8:         Get the output feature $\mathbf{y}_{t,i}^{(j)}$ base on Eq. 2;
9:     **end for**
10:     Get $\mathbf{y}_t^{(j)}$ by aggregating all $\mathbf{y}_{t,i}^{(j)}$ base on Eq. 3 for the next block;
11:     **if** $(j+1)$-th block is down-sampling block **then**
12:         $\mathbf{x}_t^{(j+1)} \leftarrow \mathbf{y}_t^{(j)}$ and push $\mathbf{y}_t^{(j)}$ into $\mathcal{S}$;
13:     **else if** $(j+1)$-th block is up-sampling block **then**
14:         Pop $\mathbf{y}_t^s$ from $\mathcal{S}$ and $\mathbf{x}_t^{(j+1)} \leftarrow \text{concat}(\mathbf{y}_t^{(j)}, \mathbf{y}_t^s)$;
15:     **else**
16:         $\mathbf{x}_t^{(j+1)} \leftarrow \mathbf{y}_t^{(j)}$;
17:     **end if**
18: **end for**
19: $\hat{\epsilon}_t \leftarrow \mathbf{x}_t^{(K)}$;
20: **return** predicted noise $\hat{\epsilon}_t$.

---

## B  DETAILS ABOUT PARAMETERS

The parameters of AFA are concentrated in the SABW module. The SABW module comprises three components, which are a ResNet layer, a Transformer layer, and a convolution layer, respectively. Among them, the number of parameters in both the ResNet layer and the convolution layer varies depending on the number of ensembled models. For instance, when AFA ensembles two diffusion models, SABW contains approximately 42.425 million parameters. With each additional diffusion model ensembled, the parameter count of SABW increases by about 2.707 million. Therefore, when three diffusion models are ensembled, the total number of the trainable parameters is 45.132 million, and with four models, it rises to approximately 47.839 million. In conclusion, we estimate that AFA encompasses close to 50 million trainable parameters.

## C    DETAILS ABOUT EVALUATION PROTOCOLS

**COCO 2017** comprises 118,287 and 5,000 image-caption pairs in the test and validation sets. All the models generate images with a resolution of 256×256. We apply four metrics to evaluate the generation performance, which are Fréchet Inception Distance (FID), Inception Score (IS), CLIP-I, and CLIP-T, respectively. FID and IS are applied to the test set, while CLIP-I and CLIP-T are applied to the validation set. Both FID and IS assess the quality of generated images. Note that lower FID means better quality. CLIP-I is the similarity between the CLIP (Radford et al., 2021) images embeddings of generated images and that of images from the image-caption pairs. CLIP-T is the CLIPScore (Hessel et al., 2021) between the generated images with the captions.

**Draw Bench Prompts** contains 200 evaluation prompts. All the models generate images with a resolution of 512×512. We apply 4 evaluation metrics, which are AES, Pick Score (PS) (Kirstain et al., 2023), HPSv2 (Wu et al., 2023), and Image Reward (IR) (Xu et al., 2023), respectively. All three metrics evaluate the performance by a model that simulates human preferences. The evaluations are conducted 20 times to ensure statistical significance. And the averaged metrics are reported.

## D    EFFECT OF MORE TRAINING SAMPLES

We conduct an experiment to assess the influence of an increased number of training samples on the performance of AFA. For this purpose, we vary the number of training samples from 0 to 18,000, increasing the increment of 2,000. As depicted in Figure 8, the trend of the lines initially rises and then stabilizes after reaching 10,000 training samples. This suggests that approximately 10,000 training samples are sufficient for AFA. More training samples do not bring greater performance. It may be due to the inherent limitations in the generative capabilities of the base models.

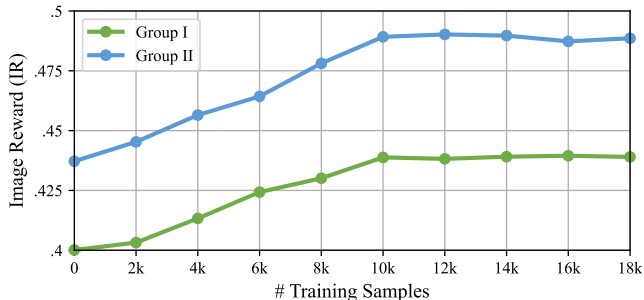

Figure 8: Effect of varying training samples.

## E    MORE QUALITATIVE COMPARISONS

More qualitative comparisons between our AFA and the base models can be found in Figure 9.

## F    ENSEMBLING BY MIXTURE-OF-EXPERTS

Another intuitive approach to ensembling multiple models involves the use of the Mixture-of-Experts (MoE) method Shen et al. (2023b); Eigen et al. (2013); Riquelme et al. (2021); Fedus et al. (2022); Du et al. (2022); Chi et al. (2022). The MoE framework incorporates multiple models, each designated as an *expert*. Given an input, MoE employs a routing mechanism to determine the most suitable expert. The input is then processed by the selected expert, and its output is considered as the output of the entire model. The number of parameters in MoE proportionally increases with the number of ensembled experts. However, the parameters of MoE are sparse, indicating that not all parameters are utilized during the inference process. Despite ensembling multiple models, only one of them is activated during inference, which makes MoE efficient.

Several text-to-image generation methods leverage MoE-based diffusion models, where multiple denoisers are ensembled along the timesteps. In these methods, only one denoising expert is activated at each timestep. For instance, ERNIE-ViLG 2.0 (Feng et al., 2023) and eDiff-I (Balaji et al.,

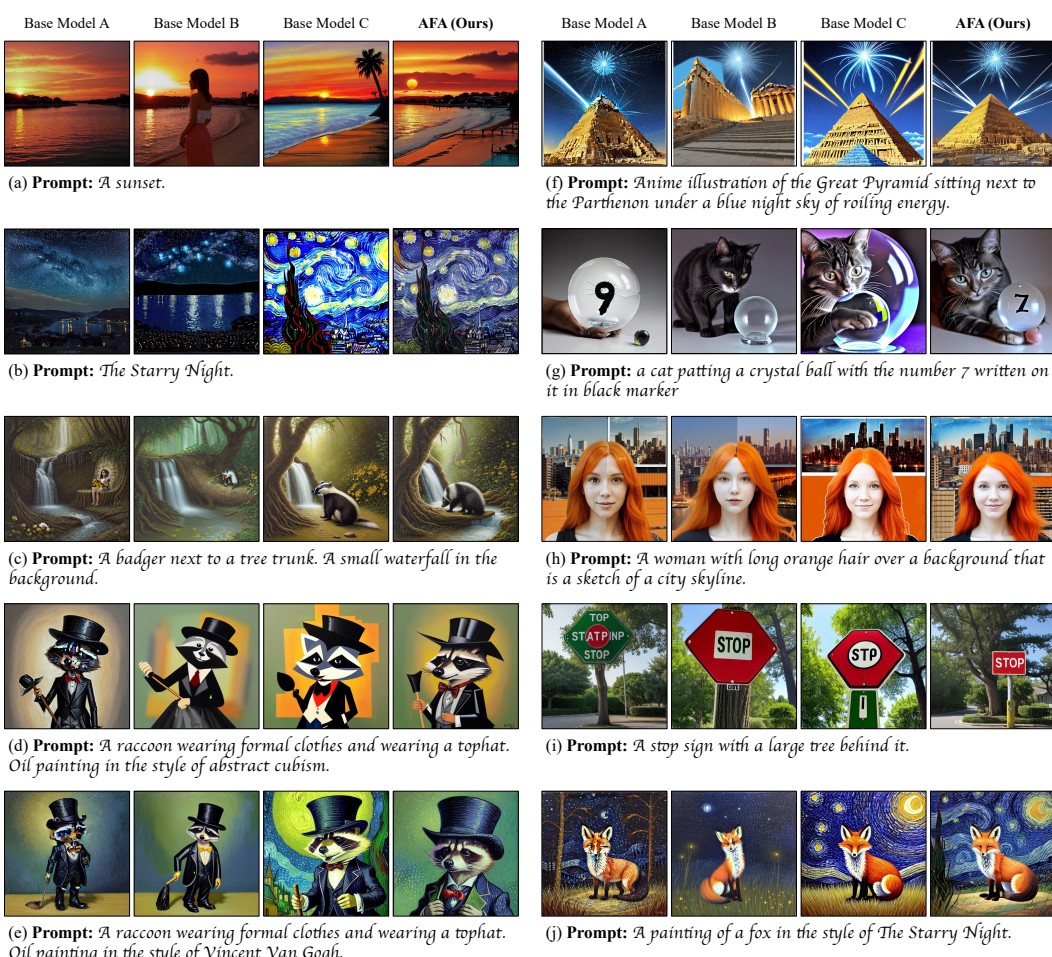

Base Model A  Base Model B  Base Model C  **AFA (Ours)**   Base Model A  Base Model B  Base Model C  **AFA (Ours)**

(a) **Prompt:** A sunset.

(f) **Prompt:** Anime illustration of the Great Pyramid sitting next to the Parthenon under a blue night sky of roiling energy.

(b) **Prompt:** The Starry Night.

(g) **Prompt:** a cat patting a crystal ball with the number 7 written on it in black marker

(c) **Prompt:** A badger next to a tree trunk. A small waterfall in the background.

(h) **Prompt:** A woman with long orange hair over a background that is a sketch of a city skyline.

(d) **Prompt:** A raccoon wearing formal clothes and wearing a tophat. Oil painting in the style of abstract cubism.

(i) **Prompt:** A stop sign with a large tree behind it.

(e) **Prompt:** A raccoon wearing formal clothes and wearing a tophat. Oil painting in the style of Vincent Van Gogh.

(j) **Prompt:** A painting of a fox in the style of The Starry Night.

Figure 9: More qualitative comparisons between our AFA and the base models.

2022) divide all the timesteps into blocks, each of which consists of consecutive timesteps and is assigned to a denoising expert. Building on this idea, MEME (Lee et al., 2024) introduces denoisers with different architectures, tailored to distinct timestep intervals. In these MoE-based methods, the denoising capability of each denoiser is restricted into some specific consecutive timesteps through large-scale full-parameters training, which is not suitable to the task of model ensembling.

Inspired by the MoE-based diffusion models discussed above, an intuitive MoE-based ensembling method is to employ a trained router to select a base denoiser for the current timesteps. We refer to this as the *Denoiser-Level MoE* method. Furthermore, to mirror the concept of our AFA, we propose another strategy of *Block-Level MoE* method, which uses a router to select the next block. The key difference between the Block-Level MoE method and our AFA lies in their operations: AFA manipulates the outputs of all blocks, whereas MoE directly acts on the input. We will introduce the details of Block-Level MoE method, as the Denoiser-Level MoE method can be considered a special case of *Block-Level MoE* method, where the selection occurs only before the first block.

Given $N$ diffusion models, we employ a router $r_\zeta^{(j)}$ to determine which model's block will be used, where $\zeta$ denotes the learnable parameters. Specifically, $r_\zeta^{(j)}$ output the logits of $N$ models,

$$\mathbf{l}_t^{(j)} = r_\zeta^{(j)}(\mathbf{x}_t^{(j)}, \mathbf{c}, t) \in \mathbb{R}^N . \tag{13}$$

During training, the probability for the $i$-th model is defined by the Gumbel softmax,

$$p_i = \frac{\exp((l_{t,i}^{(j)} + g)/\tau)}{\sum_{k=1}^{N} \exp((l_{t,k}^{(j)} + g)/\tau)} \ , \tag{14}$$

where, $l_{t,i}^{(j)}$ is the $i$-th element of $\mathbf{l}_t^{(j)}$. $g$ is the Gumbel noise. $\tau$ is the temperature coefficient. To address the issue of back-propagation incapability, we apply the reparameterization trick. Given the probability distribution $\mathbf{p} = [p_i]_{i=1}^N$, the reparameterized distribution is

$$\mathbf{p}' = \text{onehot}(\mathbf{p}) + \mathbf{p} - \text{sg}(\mathbf{p}) \ . \tag{15}$$

Among them, $\text{onehot}(\cdot)$ is the one-hot function, which sets the maximum probability to 1 and the other to 0. $\text{sg}(\cdot)$ is the stop gradient function. During the forward process, the distribution is defined as $\mathbf{p}' = \text{onehot}(\mathbf{p})$. However, during the backward process, the distribution of $\mathbf{p}' = \mathbf{p}$ is used to compute the gradients. The output of the $j$-th block is

$$\mathbf{y}_t^{(j)} = \sum \mathbf{p}' \otimes \text{concat}(\{\mathbf{y}_{t,i}^{(j)}\}_{i=1}^N) \ . \tag{16}$$

From Eq. 16, we can infer that the forward time is proportional to the number of ensembled models during training. The training loss is the denoising loss.

During inference, only the model that achieves maximum logits will be used to process the block input,

$$\mathbf{y}_t^{(j)} = \boldsymbol{\epsilon}_{\theta_k}^{(j)}(\mathbf{x}_t^{(j)}, \mathbf{c}, t) \ , \quad k = \arg\max_{i \in [N]} l_{t,i}^{(j)} \ . \tag{17}$$

The forward time of the ensembled model is the same as that of a single model during inference.

To implement the router $r_\zeta^{(j)}$, we apply a ResNet layer to introduce timesteps by adding the time embedding into the feature, and a Transformer layer to introduce textual prompts by the cross-attention. Specifically, given the input for the $j$-th block (i.e., $\mathbf{x}_t^{(j)} \in \mathbb{R}^{h_j \times w_j \times c_j}$), the output logits for $N$ models (i.e,, $\mathbf{l}_t^{(j)}$) are achieved by

$$\mathbf{h}_t^{(j)} \leftarrow \text{ResLayer}(\mathbf{x}_t^{(j)} + \gamma(t)) \in \mathbb{R}^{h_j \times w_j \times d} \ , \tag{18}$$

$$\mathbf{o}_t^{(j)} \leftarrow \text{TransformerLayer}(\mathbf{h}_t^{(j)}, \mathbf{c}) \in \mathbb{R}^{h_j \times w_j \times d} \ , \tag{19}$$

$$\mathbf{l}_t^{(j)} \leftarrow \text{Linear}(\text{AvgPool}(\mathbf{o}_t^{(j)})) \in \mathbb{R}^N \ . \tag{20}$$

Although the time required for a single forward pass in the MoE methods is approximately equivalent to that of a single model, they face challenges in performing parallel computations when generating multiple images simultaneously. This limitation becomes evident during classifier-free guidance. If the router selects two different models to process the conditioned and unconditioned inputs, MoE can only execute two sequential forward passes, rather than concurrently processing these two scenarios within a single forward pass. Moreover, when generating $K$ images at once, the MoE method has a high probability of necessitating $2K$ forward passes. While these $2K$ inputs can be divided among $N$ models, this only reduces the complexity associated with the number of forward passes from $O(K)$ to $O(N)$. This reduction still represents a substantial computational demand.

The quantitative comparison among the base models, the MoE methods, and our AFA is summarized in Table 4. AFA demonstrates superior performance compared to the MoE methods. This may be because the base denoisers have well-balanced denoising capabilities across all timesteps, and selecting a specific denoiser for a given timestep does not significantly enhance the overall denoising capability of the ensembled model. Furthermore, we compare AFA and the MoE methods using fewer inference steps (i.e., 20 steps). As shown in Table 5, unlike our AFA, the MoE methods do not demonstrate robustness with fewer inference steps, performing similarity to the base models.

The efficiency comparison among the base models, the MoE methods, and our AFA is summarized in Table 6. Firstly, the MoE methods have a similar number of parameters and training TFLOPs as our AFA. This is because the MoE methods still require all base model to be available for selection, and during training, each base model must perform a forward pass. Secondly, during inference, the MoE methods exhibit lower TFLOPs compared to AFA, as only one base model is activated at each

timestep. However, despite this advantage in TFLOPs, the inference time of the MoE methods is comparable to that of AFA due to their inability to leverage parallel inference effectively. Moreover, the MoE methods do not support fewer inference steps. In contrast, when AFA reduces the number of inference steps to 20, it still outperforms the MoE methods, even when they use 50 inference steps, while also achieving lower TFLOPs and faster inference. This highlights the efficiency and effectiveness of our AFA.

| | Group I | | | | Group II | | | |
|---|---|---|---|---|---|---|---|---|
| | AES | PS | HPSv2 | IR | AES | PS | HPSv2 | IR |
| Base Model A | 5.4102 | 21.6279 | 27.8007 | .3544 | 5.5641 | 21.8013 | 28.0183 | .4238 |
| Base Model B | 5.5013 | 21.4624 | 27.7246 | .2835 | 5.5027 | 21.7249 | 28.0343 | .4202 |
| Base Model C | 5.4881 | 21.8031 | 27.9652 | .3922 | 5.5712 | 21.4936 | 27.8089 | .3367 |
| Denoiser-Level MoE | 5.4727 | 21.6893 | 27.5345 | .4019 | 5.5638 | 21.5531 | 27.7843 | .4380 |
| Block-Level MoE | 5.5037 | 21.7690 | 27.8749 | .4037 | 5.5718 | 21.7636 | 27.9842 | .4463 |
| **AFA (Ours)** | **5.5201** | **21.8263** | **27.9734** | **.4388** | **5.5798** | **21.8059** | **28.0371** | **.4892** |

Table 4: Quantitative comparison between AFA with the base models and the MoE methods.

| | Group I | | | | Group II | | | |
|---|---|---|---|---|---|---|---|---|
| | AES | PS | HPSv2 | IR | AES | PS | HPSv2 | IR |
| Denoiser-Level MoE | 1.2310 | 2.9435 | 4.1739 | .0184 | 2.4556 | 3.6723 | 5.3415 | .0237 |
| Block-Level MoE | 1.6854 | 3.4573 | 5.3583 | .0098 | 2.5734 | 4.2368 | 4.7252 | .0344 |
| **AFA (Ours)** | **5.4951** | **21.8093** | **27.7347** | **.4191** | **5.5322** | **21.7830** | **27.9775** | **.4803** |

Table 5: Quantitative comparison of AFA and the MoE methods with 20 inference steps.

| | | | 1 image | | 2 images | | 4 images | |
|---|---|---|---|---|---|---|---|---|
| | # params.[†] | T-TLOPs | I-TFLOPs | Times (s)[‡] | I-TFLOPs | Times (s)[‡] | I-TFLOPs | Times (s)[‡] |
| Base Model[*] | 859.52M | – | 70.24 | 2.94 | 140.45 | 5.61 | 280.88 | 6.87 |
| Denoiser-Level MoE[*] | 2581.01M | 6.36 | 72.37 | 5.36 | 144.75 | 10.37 | 289.48 | 20.81 |
| Block-Level MoE[*] | 2621.03M | 6.43 | 74.41 | 5.39 | 148.82 | 10.43 | 297.65 | 21.00 |
| AFA (50 inf. steps) | 2621.03M | 6.42 | 218.74 | 8.98 | 437.48 | 15.57 | 874.96 | 21.33 |
| AFA (20 inf. steps) | 2621.03M | 6.42 | 86.61 | 3.62 | 173.22 | 6.41 | 346.45 | 8.74 |

Table 6: Efficiency Comparison between AFA with the base models and the MoE methods when ensembling three base models to generate images with a resolution of 512×512. T-TFLOPs and I-TFLOPs denote TFLOPs for training and inference, respectively. [†]Note that only the parameters of denoisers are considered. [‡]Note that all the inference times are evaluated in our A6000 environment over 20 runs. [*]Note that both the base models and MoE methods are evaluated with 50 inference steps, because they cannot achieve comparable performance with fewer inference steps.

# G  ENSEMBLING MORE MODELS

To evaluate the scalability of our AFA, we apply it to ensemble a larger number of base models. Specifically, we select and ensemble several base models with best performance, and test IR metrics in Draw Bench Prompts. As shown in Table 7, ensembling more base models leads more performance improvements. Additionally, increasing the number of the ensembled base models enhances the tolerance for fewer inference steps.

# H  COMPARISON ON FEWER INFERENCE STEPS

We compare our AFA with the baseline methods under fewer inference steps. As shown in Table 8, our AFA not only outperforms the baseline methods at higher inference steps but also demonstrates significantly better performance under fewer inference steps. This highlights AFA's superior tolerance to the fewer inference steps compared to the baselines.

| | 1 base model | 2 base models | 3 base models | 4 base models | 5 base models | 6 base models |
|---|---|---|---|---|---|---|
| 50 inference steps | 0.4238 | 0.5003 | 0.4892 | 0.4967 | 0.5042 | 0.5347 |
| 40 inference steps | 0.4065 | 0.4834 | 0.4874 | 0.4906 | 0.4984 | 0.5310 |
| 30 inference steps | 0.3593 | 0.4791 | 0.4846 | 0.4853 | 0.4993 | 0.5294 |
| 20 inference steps | 0.2103 | 0.3641 | 0.4803 | 0.4840 | 0.4975 | 0.5304 |
| 10 inference steps | -0.5255 | 0.1844 | 0.4013 | 0.4285 | 0.4423 | 0.5135 |

Table 7: Performance (IR) comparison across different numbers of ensembled base models and varying inference steps.

| | Group I | | | | | Group II | | | | |
|---|---|---|---|---|---|---|---|---|---|---|
| inference steps | 50 | 40 | 30 | 20 | 10 | 50 | 40 | 30 | 20 | 10 |
| Wtd. Merging | 0.3909 | 0.3104 | 0.2451 | -0.0535 | -0.7536 | 0.4387 | 0.4123 | 0.2942 | 0.1046 | -0.5105 |
| MBW | 0.3922 | 0.3041 | 0.2345 | 0.0341 | -0.7731 | 0.4396 | 0.4094 | 0.2893 | -0.0031 | -0.5524 |
| autoMBW | 0.3672 | 0.3158 | 0.2510 | -0.4173 | -0.8046 | 0.3513 | 0.3545 | 0.2502 | -0.0841 | -0.5841 |
| MagicFusion | 0.3317 | 0.3245 | 0.3104 | 0.2349 | 0.0175 | 0.4194 | 0.3995 | 0.3408 | 0.1951 | 0.0818 |
| AFA (Ours) | **0.4388** | **0.4389** | **0.4238** | **0.4191** | **0.3575** | **0.4892** | **0.4874** | **0.4846** | **0.4803** | **0.4013** |

Table 8: Quantitative comparison (IR) on fewer inference steps.

## I  EFFICIENCY COMPARISON

We compare the efficiency of merging methods and ensembling methods. As shown in Table 9, when using the same 50 inference steps, ensembling methods (e.g., MagicFusion and AFA) do not have an efficiency advantage over a single base models or merging methods. The number of model parameters, TFLOPs, and inference time all increase linearly with the number of base models.

However, thanks to AFA's high tolerance for fewer inference steps, it can achieve similar performance with reduced steps, as demonstrated in Table 7, while the baseline methods fail to maintain performance, as shown in Table 8. Therefore, under fewer inference steps, AFA achieves comparable efficiency (TFLOPs and inference time) to that of a single base models and merging methods.

| | 2 base models | | | 3 base models | | | 6 base models | | |
|---|---|---|---|---|---|---|---|---|---|
| | # params.[†] | TFLOPs | Times (s)[‡] | # params.[†] | TFLOPs | Times (s)[‡] | # params.[†] | TFLOPs | Times (s)[‡] |
| Merging Method[*] | 859.52M | 70.24 | 2.94 | 859.52M | 70.24 | 2.94 | 859.52M | 70.24 | 2.94 |
| MagicFusion[*] | 1719.04M | 138.66 | 6.01 | 2578.56M | 206.88 | 9.13 | 5157.12M | 411.53 | 22.41 |
| AFA (50 inf. steps) | 1761.47M | 148.64 | 6.28 | 2621.03M | 218.74 | 8.98 | 5210.37M | 430.08 | 23.15 |
| AFA (30 inf. steps) | 1761.47M | 94.24 | 3.21 | – | – | – | – | – | – |
| AFA (20 inf. steps) | – | – | – | 2621.03M | 86.61 | 3.62 | – | – | – |
| AFA (10 inf. steps) | – | – | – | – | – | – | 5210.37M | 103.68 | 4.58 |

Table 9: Efficiency Comparison between the merging methods and the ensembling methods when generating an image with a resolution of 512×512. [†]Note that only the parameters of denoisers are considered. [‡]Note that all the inference times are evaluated in our A6000 environment over 20 runs. [*]Note that both the merging methods and MagicFusion are evaluated with 50 inference steps, because they cannot achieve comparable performance with fewer inference steps.

## J  GENERALITY ON OTHER ARCHITECTURES

To evaluate the generality of our AFA on other diffusion architectures, we select three SDXL (Podell et al., 2023) models and two FLUX .1 [dev][16] models from CivitAI.

SDXL consists two U-Nets: the denoising U-Net and the refining U-Net. Both U-Nets shares a similar architecture of SDv1.5, but contain larger blocks. Consistent with the method used for ensembling SDv1.5 models, we employ the SABW module to aggregate features from each block.

---

[16]https://blackforestlabs.ai/announcing-black-forest-labs/

FLUX .1 [dev] is a DiT-based model (Peebles & Xie, 2023). To ensemble models with this architecture, we also utilize the SABW module to aggregate features from each DiT Transformer block.

The selected SDXL models are NoobAI-XL[17], iNiverse-Mix[18], and epiCRealism-XL[19]. The selected FLUX .1 [dev] models are PixelWave[20] and VerusVision[21].

As shown in Table 10, when ensembling models with the SDXL or FLUX .1 [dev] architectures, the performance of the ensembled model surpasses that of the individual base models, demonstrating the generality of our AFA on different model architectures. However, the observed improvement is relatively small, likely because the base models already exhibit strong performance, and ensembling leads to only marginal gains. Additionally, due to the large size of the base models, ensembling multiple such models may be less practical.

It is worth noting that our AFA cannot be applied to ensembling models with different architectures, as the misalignment of block features makes block-wise aggregation challenging. We view ensembling models with diverse architectures as a promising direction for future research.

|  | SDXL | | | | FLUX .1 [dev] | | | |
|---|---|---|---|---|---|---|---|---|
|  | AES | PS | HPSv2 | IR | AES | PS | HPSv2 | IR |
| Base Model A | 6.1948 | 21.7854 | 28.0452 | .6314 | 6.2341 | 21.8843 | 28.0958 | .7341 |
| Base Model B | 6.1363 | 21.7783 | 28.0531 | .6328 | 6.2346 | 21.8593 | 28.1003 | .7324 |
| Base Model C | 6.1852 | 21.7752 | 27.9984 | .6339 | – | – | – | – |
| **AFA (Ours)** | **6.2190** | **21.7959** | **28.0894** | **.6342** | **6.2490** | **21.8931** | **28.1194** | **.7370** |

Table 10: Quantitative comparison of AFA with different diffusion architectures.

## K    DISTILLATION TO FEWER INFERENCE STEPS

While our AFA demonstrates robustness to fewer inference steps, we aim to further distill the ensembled model into significantly fewer steps. Specifically, we explore using LCM (Luo et al., 2023) to achieve this. In our experiments, we kept the parameters of the denoisers frozen while training only the parameters of the SABW modules. However, as shown in Table 11, this approach is unsuccessful. A possible reason for this failure is that the frozen parameters may have impeded the distillation process.

|  | Original Ensembled Model | | Distilled Ensembled Model | | | |
|---|---|---|---|---|---|---|
| Inference Steps | 50 | 20 | 20 | 4 | 2 | 1 |
| IR | 0.4892 | 0.4803 | 0.4793 | -0.5846 | -0.5594 | -0.5952 |

Table 11: Experimental results (IR) for distilling the ensembled model with AFA into fewer inference steps using LCM.

## L    EVALUATION ON MORE DATASETS

To validate the generality of our AFA, we evaluate it on additional datasets, which are DiffusionDB (Wang et al., 2022), JourneyDB (Pan et al., 2023), and LAION-COCO[22], respectively. We randomly selected 50,000 samples from each dataset to compare the performance of AFA against the base models and the baseline methods[23].

---

[17] https://civitai.com/models/833294?modelVersionId=1070239
[18] https://civitai.com/models/226533?modelVersionId=608842
[19] https://civitai.com/models/277058?modelVersionId=1074830
[20] https://civitai.com/models/141592?modelVersionId=992642
[21] https://civitai.com/models/883426?modelVersionId=988886
[22] https://laion.ai/blog/laion-coco/
[23] Note that for JourneyDB, we ensured the selected samples are distinct from those in the training set.

The quantitative comparisons on the three additional datasets are presented in Table 12, Table 13, and Table 14, respectively. The results demonstrate that our AFA consistently outperforms both the base models and the baseline methods, highlighting the generality and effectiveness of our approach.

| | Group I | | | | | | | | Group II | | | | | | | |
|---|---|---|---|---|---|---|---|---|---|---|---|---|---|---|---|---|
| | FID ↓ | IS | CLIP-I | CLIP-T | AES | PS | HPSv2 | IR | FID ↓ | IS | CLIP-I | CLIP-T | AES | PS | HPSv2 | IR |
| Base Model A | 14.78 | 6.88 | .6433 | .2987 | 5.557 | 23.4113 | 29.9126 | .4354 | 14.31 | 7.02 | .6578 | .3124 | 5.612 | 23.5527 | 30.1521 | .4413 |
| Base Model B | 15.01 | 6.93 | .6513 | .3010 | 5.456 | 23.1123 | 29.8131 | .4223 | 14.98 | 7.01 | .6618 | .3128 | 5.502 | 23.3128 | 30.0035 | .4307 |
| Base Model C | 15.13 | 6.87 | .6441 | .2988 | 5.478 | 23.3423 | 29.9341 | .4339 | 15.02 | 6.94 | .6391 | .2983 | 5.481 | 23.3420 | 29.9413 | .4291 |
| Wtd. Merging | 17.41 | 6.37 | .6214 | .2776 | 5.213 | 23.2975 | 28.9938 | .4007 | 16.98 | 6.81 | .6231 | .2984 | 5.501 | 23.2139 | 28.7310 | .4147 |
| MBW | 16.67 | 6.81 | .6378 | .2843 | 5.367 | 23.3002 | 28.8393 | .4115 | 15.20 | 6.99 | .6392 | .2931 | 5.493 | 23.4412 | 29.0012 | .4216 |
| autoMBW | 16.66 | 6.76 | .6342 | .2663 | 5.377 | 23.3874 | 28.8432 | .4293 | 15.17 | 7.00 | .6381 | .2891 | 5.551 | 23.3584 | 29.8413 | .4298 |
| MagicFusion | 14.77 | 6.98 | .6561 | .3123 | 5.674 | 23.4132 | 29.9241 | .4440 | 14.21 | 7.02 | .6641 | .3125 | 5.611 | 23.5049 | 29.9941 | .4391 |
| **AFA (Ours)** | **12.32** | **7.39** | **.6765** | **.3691** | **5.773** | **23.5132** | **30.0946** | **.4763** | **13.98** | **7.11** | **.6712** | **.3211** | **5.698** | **23.6130** | **30.1931** | **.4712** |

Table 12: Quantitative comparison in Group I and II on DiffusionDB.

| | Group I | | | | | | | | Group II | | | | | | | |
|---|---|---|---|---|---|---|---|---|---|---|---|---|---|---|---|---|
| | FID ↓ | IS | CLIP-I | CLIP-T | AES | PS | HPSv2 | IR | FID ↓ | IS | CLIP-I | CLIP-T | AES | PS | HPSv2 | IR |
| Base Model A | 17.80 | 5.63 | .5324 | .2484 | 4.520 | 19.2066 | 24.5997 | .3606 | 17.21 | 5.75 | .5378 | .2513 | 4.577 | 19.1089 | 24.5196 | .3633 |
| Base Model B | 18.17 | 5.72 | .5379 | .2501 | 4.466 | 18.9965 | 24.5139 | .3458 | 18.01 | 5.72 | .5428 | .2588 | 4.482 | 18.9069 | 24.3972 | .3483 |
| Base Model C | 17.98 | 5.62 | .5270 | .2434 | 4.507 | 19.1876 | 24.5743 | .3554 | 17.87 | 5.61 | .5169 | .2398 | 4.445 | 18.9523 | 24.2945 | .3443 |
| Wtd. Merging | 20.98 | 5.26 | .5170 | .2314 | 4.288 | 19.1237 | 23.8558 | .3327 | 20.55 | 5.50 | .5108 | .2452 | 4.455 | 18.8782 | 23.3782 | .3399 |
| MBW | 19.74 | 5.58 | .5218 | .2365 | 4.399 | 19.1381 | 23.7068 | .3371 | 18.35 | 5.74 | .5221 | .2378 | 4.484 | 19.0703 | 23.5806 | .3465 |
| autoMBW | 20.07 | 5.54 | .5204 | .2227 | 4.429 | 19.1960 | 23.6664 | .3508 | 18.04 | 5.70 | .5182 | .2325 | 4.501 | 18.9612 | 24.2498 | .3510 |
| MagicFusion | 17.76 | 5.76 | .5412 | .2591 | 4.672 | 19.2147 | 24.5909 | .3666 | 16.93 | 5.72 | .5414 | .2573 | 4.550 | 19.0955 | 24.3816 | .3570 |
| **AFA (Ours)** | **14.86** | **6.04** | **.5587** | **.3063** | **4.763** | **19.3324** | **24.7316** | **.3849** | **16.72** | **5.78** | **.5482** | **.2646** | **4.651** | **19.1986** | **24.5480** | **.3990** |

Table 13: Quantitative comparison in Group I and II on JourneyDB.

| | Group I | | | | | | | | Group II | | | | | | | |
|---|---|---|---|---|---|---|---|---|---|---|---|---|---|---|---|---|
| | FID ↓ | IS | CLIP-I | CLIP-T | AES | PS | HPSv2 | IR | FID ↓ | IS | CLIP-I | CLIP-T | AES | PS | HPSv2 | IR |
| Base Model A | 13.82 | 7.82 | .7328 | .3432 | 6.260 | 26.5012 | 33.8765 | .4962 | 13.26 | 8.06 | .7549 | .3559 | 6.402 | 26.8726 | 34.4362 | .5080 |
| Base Model B | 14.12 | 7.89 | .7397 | .3467 | 6.164 | 26.1997 | 33.7818 | .4754 | 13.87 | 8.05 | .7583 | .3617 | 6.286 | 26.5950 | 34.2635 | .4910 |
| Base Model C | 14.08 | 7.73 | .7280 | .3371 | 6.197 | 26.4257 | 33.8805 | .4886 | 13.84 | 7.90 | .7284 | .3379 | 6.254 | 26.6099 | 34.1549 | .4879 |
| Wtd. Merging | 16.46 | 7.24 | .7101 | .3209 | 5.886 | 26.3613 | 32.8551 | .4573 | 15.79 | 7.76 | .7149 | .3415 | 6.271 | 26.5130 | 32.8277 | .4762 |
| MBW | 15.45 | 7.68 | .7243 | .3275 | 6.071 | 26.3787 | 32.6951 | .4688 | 14.11 | 8.07 | .7344 | .3327 | 6.311 | 26.7870 | 33.1185 | .4860 |
| autoMBW | 15.61 | 7.67 | .7218 | .3101 | 6.092 | 26.4808 | 32.6464 | .4852 | 13.98 | 8.00 | .7328 | .3280 | 6.343 | 26.6503 | 34.0794 | .4946 |
| MagicFusion | 13.80 | 7.92 | .7477 | .3594 | 6.425 | 26.5094 | 33.8799 | .5048 | 13.13 | 8.05 | .7632 | .3632 | 6.421 | 26.8431 | 34.2797 | .5024 |
| **AFA (Ours)** | **11.66** | **8.36** | **.7716** | **.4241** | **6.582** | **26.6673** | **34.1098** | **.5331** | **12.84** | **8.16** | **.7717** | **.3735** | **6.552** | **26.9677** | **34.4844** | **.5405** |

Table 14: Quantitative comparison in Group I and II on LAION-COCO.

## M    ENSEMBLING MODELS WITH HIGHLY CORRELATED FEATURES

As the number of the base models increases, reliance on multiple base models may leads to overfitting to correlated features. To evaluate the robustness of our AFA against highly correlated features, we design an experiment. Specifically, we selected one high-quality base model and one low-quality base model. Using AFA, we ensemble the high-quality model with several low-quality models to evaluate whether the ensembled model will perform toward that of the low-quality model due to the dominance of the correlated features from the low-quality model.

As shown in Table 15, although the performance the ensembled model slightly declines when ensembling with large number of low-quality models, it still outperforms the high-quality model. This demonstrates that our AFA exhibits strong robustness to highly correlated features.

## N    TRAINING WITH DATASET OF LOWER QUALITY

In this paper, we train our AFA using a high-quality dataset, JourneyDB. This raises the concern of whether AFA's high performance is primarily attributed to the quality of the training dataset.

| | AES | PS | HPSv2 | IR |
|---|---|---|---|---|
| High-Quality Base Model | 5.5641 | 21.8013 | 28.0183 | .4238 |
| Low-Quality Base Model | 5.5013 | 21.4624 | 27.7246 | .2835 |
| AFA (1 High-Quality Model + 1 Low-Quality Model) | 5.6013 | 21.9341 | 28.3485 | .4746 |
| AFA (1 High-Quality Model + 2 Low-Quality Models) | 5.6000 | 21.9348 | 28.3398 | .4739 |
| AFA (1 High-Quality Model + 3 Low-Quality Models) | 5.5974 | 21.8248 | 28.3399 | .4730 |
| AFA (1 High-Quality Model + 5 Low-Quality Models) | 5.5831 | 21.8019 | 28.2035 | .4593 |

Table 15: Quantitative comparison of ensembling models with high-correlated features.

To address this, we also train AFA using LAION-COCO, a dataset of lower quality compared to JourneyDB.

As shown in Table 16, the model trained on LAION-COCO performs similarly to the one trained on JourneyDB, indicating that AFA is not sensitive to the quality of the training dataset. Its consistent performance across datasets of varying quality highlights its robustness and generalization ability. Regardless of whether the training dataset is high-quality or low-quality, AFA effectively leverages the features of the base models to deliver stable and reliable results.

| | Group I | | | | | | | Group II | | | | | | |
|---|---|---|---|---|---|---|---|---|---|---|---|---|---|---|
| | FID ↓ | IS | CLIP-I | CLIP-T | AES | PS | HPSv2 | IR | FID ↓ | IS | CLIP-I | CLIP-T | AES | PS | HPSv2 | IR |
| JourneyDB | 9.76 | 7.14 | .6926 | .2675 | 5.5201 | 21.8263 | 27.9734 | .4388 | 10.27 | 7.42 | .6855 | .2717 | 5.5798 | 21.8059 | 28.0371 | .4892 |
| LAION-COCO | 9.74 | 7.16 | .6930 | .2664 | 5.5093 | 21.8257 | 27.9781 | .4390 | 10.29 | 7.39 | .6861 | .2711 | 5.5789 | 21.8077 | 28.0363 | .4889 |

Table 16: Quantitative comparison in Group I and II trained on JourneyDB and LAION-COCO.

## O  MORE EXPERIMENTAL DETAILS

We train our AFA in an environment equipped with 8 NVIDIA V100 GPUs, each with 32GB of memory. When ensembling two base models, the GPU memory consumption is about 11GB under FP16 precision and a batch size of 1. When ensembling three base models, the memory consumption increases to 14GB under the same settings. For six base models, the memory consumption reaches about 30GB. Due to the frozen parameters of the base models, training our AFA demands relatively modest GPU memory, even when ensembling a larger number of base models.

