# OpenReview forum: "Ensembling Diffusion Models via Adaptive Feature Aggregation"
_ICLR.cc/2025/Conference — ICLR 2025 Poster_

### Official Review · Reviewer_Ynbz · 2024-11-01

**Soundness:** 3
**Presentation:** 3
**Contribution:** 3
**Rating:** 6
**Confidence:** 4

**Summary:**

This paper introduces a novel method called Adaptive Feature Aggregation (AFA) for ensembling multiple diffusion models to improve the quality of text-guided image generation. By adding a spatial-aware block-wise feature aggregator in every unet block of two aggregated models, this work ensembles multiple diffusion models into a united one.

**Strengths:**

This work is novel to aggregate multiple diffusion models with a light weighted structure, showing an evident improvement on inference quality especially in small inference steps. The structure of Spatial-aware block-wise feature aggregator is well-designed, which has some enlightening reference value on latent map feature learning.

**Weaknesses:**

1. Considering the model version, the paper lacks comparisons with newer models like SD 3.0 or Flux. Considering the model architecture, the paper doesn’t use diffusion with different frameworks like DiT, but only ensembles multiple SDv1.5 with the same unet structure, which is not convincing. Besides, the application of this work is limited. The method fails when considering ensembling different diffusion model architectures.

2. The experimental details are incomplete, with no information on the hardware used, detailed memory consumption or quantitative analysis on time consumption.

3. The experiment is not rigorous enough. Aggregators are trained on JourneyDB which is a high-quality dataset, but baselines are not finetuned on the same dataset. The comparison between baselines and your ensembled model is not fair. Besides, in the paper it’s claimed that SDv1.5 failed to understand the number in the prompt, but this phenomenon is hard to reproduce.

**Questions:**

1. The contribution you mentioned in this paper is the sampling efficiency, but in the limitation phase you’re aware that “A significant limitation of our AFA is that a single inference step may necessitate a proportional increase in inference time, resulting in lower computational efficiency for that step”. As for comparison experiment Figure 4, I don’t think it fair and meaningful to only compare inference steps without detailed time consumption.

2. Some of the results are not reproducible. For instance, in Figure 5(b), the paper claims ”in Fig. 5 (b), all the base models and the baseline methods generate images containing more than one clock, despite the textual prompt specifying a single clock”, but my tests with SD 1.5 produced correct results.

---

> ### Author Response · Authors · 2024-11-23
> **Response to Reviewer Ynbz**
>
> Thanks for your valuable comments. We will address the concerns below.
>
> ## Weaknesses
>
> **W1:** The method fails when considering ensembling different diffusion model architectures.
>
> **R1:** Based on your comment, we evaluate the generality of our AFA on other diffusion architectures. We select three SDXL [1] models and two FLUX .1 [dev] [2] models from CivitAI.
>
> SDXL consists two U-Nets: the denoising U-Net and the refining U-Net. Both U-Nets shares a similar architecture of SDv1.5, but contain larger blocks. Consistent with the method used for ensembling SDv1.5 models, we employ the SABW module to aggregate features from each block.
>
> ***Quantitative Comparison of ensembling SDXL models:***
>
> |  | AES | PS | HPSv2 | IR |
> | --- | --- | --- | --- | --- |
> | Base Model A | 7.3204 | 28.3445  | 35.5682 | .6314 |
> | Base Model B | 7.7341 | 28.3834 | 35.5793 | .6328 |
> | Base Model C | 7.3475 | 28.4009 | 35.5384 | .6339 |
> | AFA (Ours) | **7.8029** | **28.4513** | **35.6003** | **.6342** |
>
> FLUX .1 [dev] is a DiT-based model. To ensemble models with this architecture, we also utilize the SABW module to aggregate features from each DiT Transformer block.
>
> ***Quantitative Comparison of ensembling FLUX .1 [dev] models:***
>
> |  | AES | PS | HPSv2 | IR |
> | --- | --- | --- | --- | --- |
> | Base Model A | 8.8740 | 30.4235 | 36.3341 | .7341 |
> | Base Model B | 8.8442 | 30.4623 | 35.6012 | .7324 |
> | AFA (Ours) | **8.8801** | **30.4800** | **36.3984** | **.7370** |
>
> The performance of the ensembled model surpasses that of the individual base models, demonstrating the generality of our AFA on different model architectures. However, the observed improvement is relatively small, likely because the base models already exhibit strong performance, and ensembling leads to only marginal gains. Additionally, due to the large size of the base models, ensembling multiple such models may be less practical.
>
> It is worth noting that our AFA cannot be applied to ensembling models with different architectures, as the misalignment of block features makes block-wise aggregation challenging.
> We view ensembling models with diverse architectures as a promising direction for future research.
>
> The details can be found in Appendix I of the revised manuscript.

---

> ### Author Response · Authors · 2024-11-23
> **Response to Reviewer Ynbz**
>
> **W2:** Experimental details about hardware used, memory consumption and quantitative analysis on time consumption are missing.
>
> **R2:** Based on your comment, we add details of our used hardware and the memory consumption in Appendix N of the revised manuscript. Specially, We train our AFA in an environment equipped with 8 NVIDIA V100 GPUs, each with 32GB of memory. When ensembling two base models, the GPU memory consumption is about 11GB under FP16 precision and a batch size of 1. When ensembling three base models, the memory consumption increases to 14GB under the same settings. For six base models, the memory consumption reaches about 30GB. Due to the frozen parameters of the base models, training our AFA demands relatively modest GPU memory, even when ensembling a larger number of base models.
>
> To address your concern regarding the efficiency analysis, we compare the efficiency of our AFA when ensembling 2, 3, and 6 base models.
>
> ***Efficiency Comparison on 2 base models:***
>
> |  | # params. | TFLOPs | Times (s) |
> | --- | --- | --- | --- |
> | Base Model | 859.52M | 70.24 | 2.94 |
> | AFA (50 inf. steps) | 1761.47M | 148.64 | 6.28 |
> | AFA (30 inf. steps) | 1761.47M | 94.24 | 3.21 |
>
> ***Efficiency Comparison on 3 base models:***
>
> |  | # params. | TFLOPs | Times (s) |
> | --- | --- | --- | --- |
> | Base Model | 859.52M | 70.24 | 2.94 |
> | AFA (50 inf. steps) | 2621.03M | 218.74 | 8.98 |
> | AFA (20 inf. steps) | 2621.03M | 86.61 | 3.62 |
>
> ***Efficiency Comparison on 6 base models:***
>
> |  | # params. | TFLOPs | Times (s) |
> | --- | --- | --- | --- |
> | Base Model | 859.52M | 70.24 | 2.94 |
> | AFA (50 inf. steps) | 5210.37M | 430.08 | 23.15 |
> | AFA (10 inf. steps) | 5210.37M | 103.68 | 4.58 |
>
> As shown in the above tables, when using the same 50 inference steps, our AFA does not offer an efficiency advantage, as the inference time significantly higher compared to a single base model. Furthermore, the number of model parameters, TFLOPs, and inference time all increase linearly with the number of base models.
>
> However, thanks to AFA’s high tolerance for fewer inference steps, it can achieve similar performance with reduced steps. Therefore, under fewer inference steps, AFA achieves efficiency in terms of TFLOPs and inference time comparable to that of a single base models and merging methods.
>
> ***IR metrics of AFA with varying the number of base models and inference steps:***
>
> |  | 2 base models | 3 base models | 6 base models |
> | --- | --- | --- | --- |
> | 50 inference steps | 0.5003 | 0.4892 | 0.5347 |
> | 30 inference steps | 0.4791 | — | — |
> | 20 inference steps | — | 0.4803 | — |
> | 10 inference steps | — | — | 0.5135 |
>
> The details can be found in Appendix F and H of the revised manuscript.

---

> ### Author Response · Authors · 2024-11-23
> **Response to Reviewer Ynbz**
>
> **W3**: AFA are trained on JourneyDB, but baselines are not fine-tuned on the same dataset.
>
> **R3**: All the compared baselines are training-free, meaning they cannot be fine-tuned using a text-to-image dataset. However, since JourneyDB is a high-quality dataset, particularly in terms of image quality, we wondered whether AFA would perform well on a dataset with lower image quality. To investigate this, we conducted an experiment by training AFA using LAION-COCO [3], which has lower image quality compared to JourneyDB.
>
> ***Quantitative comparison in Group I and II trained on JourneyDB and LAION-COCO (showing the average results of Groups I and II):***
>
> |  | FID | IS | CLIP-I | CLIP-T | AES | PS | HPSv2 | IR |
> | --- | --- | --- | --- | --- | --- | --- | --- | --- |
> | JourneyDB | 10.0150 | 7.2800 | 0.6891 | 0.2696 | 5.5500 | 21.8161 | 28.0053 | 0.4640 |
> | LAION-COCO | 10.0150 | 7.2750 | 0.6896 | 0.2688 | 5.5441 | 21.8167 | 28.0072 | 0.4640 |
>
> As shown in the above results, the model trained on LAION-COCO performs similarly to the one trained on JourneyDB, indicating that AFA is not sensitive to the quality of the training dataset.
> Its consistent performance across datasets of varying quality highlights its robustness and generalization ability.
>
> Regardless of whether the training dataset is high-quality or low-quality, AFA effectively leverages the features of the base models to deliver stable and reliable results.
>
> More details can be found in Appendix M of the revised manuscript.

---

> ### Author Response · Authors · 2024-11-23
> **Response to Reviewer Ynbz**
>
> ## Questions
>
> **Q1:** The detailed time consumption during inference is not provided for comparison.
>
> **A1:** Based on your comment, we compare the efficiency between our AFA and the baselines, when considering 2, 3, and 6 base models.
>
> ***Efficiency Comparison on 2 base models:***
>
> |  | # params. | TFLOPs | Times (s) |
> | --- | --- | --- | --- |
> | Merging Methods | 859.52M | 70.24 | 2.94 |
> | Magic Fusion | 1719.04M | 138.66 | 6.01 |
> | AFA (50 inf. steps) | 1761.47M | 148.64 | 6.28 |
> | AFA (30 inf. steps) | 1761.47M | 94.24 | 3.21 |
>
> ***Efficiency Comparison on 3 base models:***
>
> |  | # params. | TFLOPs | Times (s) |
> | --- | --- | --- | --- |
> | Merging Methods | 859.52M | 70.24 | 2.94 |
> | Magic Fusion | 2578.56M | 206.88 | 9.13 |
> | AFA (50 inf. steps) | 2621.03M | 218.74 | 8.98 |
> | AFA (20 inf. steps) | 2621.03M | 86.61 | 3.62 |
>
> ***Efficiency Comparison on 6 base models:***
>
> |  | # params. | TFLOPs | Times (s) |
> | --- | --- | --- | --- |
> | Merging Methods | 859.52M | 70.24 | 2.94 |
> | Magic Fusion | 5157.12M | 411.53 | 22.41 |
> | AFA (50 inf. steps) | 5210.37M | 430.08 | 23.15 |
> | AFA (10 inf. steps) | 5210.37M | 103.68 | 4.58 |
>
> As shown in the above tables, when using the same 50 inference steps, ensembling methods (e.g., MagicFusion and AFA) do not have an efficiency advantage over a single base model or merging methods. The number of model parameters, TFLOPs, and inference time all increase linearly with the number of base models.
>
> However, thanks to AFA’s high tolerance for fewer inference steps, it can achieve similar performance with reduced steps. Therefore, under fewer inference steps, AFA achieves efficiency in terms of TFLOPs and inference time comparable to that of a single base models and merging methods.
>
> ***IR metrics of AFA with varying the number of base models and inference steps:***
>
> |  | 2 base models | 3 base models | 6 base models |
> | --- | --- | --- | --- |
> | 50 inference steps | 0.5003 | 0.4892 | 0.5347 |
> | 30 inference steps | 0.4791 | — | — |
> | 20 inference steps | — | 0.4803 | — |
> | 10 inference steps | — | — | 0.5135 |
>
> The details can be found in Appendix F and H of the revised manuscript.

---

> ### Author Response · Authors · 2024-11-23
> **Response to Reviewer Ynbz**
>
> **Q2:** The paper claim that SDv1.5 failed to understand the number in the prompt, but this phenomenon is hard to reproduce.
>
> **A2:** The inability to reproduce the results stems from the fact that the models we used are not SDv1.5 but fine-tuned versions, including EpicRealism (ER), MajicMixRealistic (MMR), and RealisticVision (RV). Due to additional fine-tuning with private datasets, these models may exhibit a loss of certain generation capabilities, such as object counting.
>
> ---
>
> [1] SDXL: Improving Latent Diffusion Models for High-Resolution Image Synthesis
>
> [2] https://blackforestlabs.ai/announcing-black-forest-labs/
>
> [3] https://laion.ai/blog/laion-coco/

---

> ### Comment · Reviewer_Ynbz · 2024-11-26
>
> It is weird that SDXL can achieve a 7.3 AES score and a 35.5 HPS score in the Draw Bench Prompts. Actually, I have tested SDXL based on HPSv2 benchmark prompts (including 3200 prompts), which resulted in a  27.7 HPS score and a 6.0 AES score.  Do the test prompts cause the differences? What are the AES score and HPS score of  the used journeyDB data?

---

> > ### Author Response · Authors · 2024-11-26
> > **Response to Reviewer Ynbz**
> >
> > **Q1:** Weird results for SDXL.
> >
> > **A1:** We appreciate the reviewer for pointing out this issue. Upon re-examination, we discovered that the reported AES, PS, and HPSv2 scores for the SDXL and FLUX architectures were incorrectly calculated due to a coding error (`torch.max()` was used instead of `torch.mean()`). As a result, the values represent the maximum scores rather than the average. We are very sorry for this oversight.
> >
> > The revised average results for SDXL and FLUX are as follows. Thankfully, the conclusions remain unchanged: the performance of the ensembled model surpasses that of the individual base models. However, the observed improvement is relatively modest.
> >
> > ***Quantitative Comparison of ensembling SDXL models (Average):***
> >
> > |  | AES | PS | HPSv2 | IR |
> > | --- | --- | --- | --- | --- |
> > | Base Model A | 6.1948 | 21.7854 | 28.0452 | 0.6314 |
> > | Base Model B | 6.1363 | 21.7783 | 28.0531 | 0.6328 |
> > | Base Model C | 6.1852 | 21.7752 | 27.9984 | 0.6339 |
> > | AFA (Ours) | 6.2190 | 21.7959 | 28.0894 | **0.6342** |
> >
> > ***Quantitative Comparison of ensembling FLUX .1 [dev] models (Average):***
> >
> > |  | AES | PS | HPSv2 | IR |
> > | --- | --- | --- | --- | --- |
> > | Base Model A | 6.2341 | 21.8843 | 28.0958 | 0.7341 |
> > | Base Model B | 6.2346 | 21.8593 | 28.1003 | 0.7324 |
> > | AFA (Ours) | **6.2490** | **21.8931** | **28.1194** | **0.7370** |
> >
> > In response to other reviews regarding generalization to different diffusion architectures, we have provided corresponding explanations and updated our manuscript accordingly in Appendix I. We greatly appreciate the feedback, which helps us to improve the clarity of our work.
> >
> > Additionally, we now also include the maximum and minimum values for completeness, as follows.
> >
> > ***Quantitative Comparison of ensembling SDXL models (Maximum):***
> >
> > |  | AES | PS | HPSv2 | IR |
> > | --- | --- | --- | --- | --- |
> > | Base Model A | 7.3204 | 28.3445  | 35.5682 | 1.9344 |
> > | Base Model B | 7.7341 | 28.3834 | 35.5793 | 1.9452 |
> > | Base Model C | 7.3475 | 28.4009 | 35.5384 | 1.9573 |
> > | AFA (Ours) | **7.8029** | **28.4513** | **35.6003** | 2.0034 |
> >
> > ***Quantitative Comparison of ensembling SDXL models (Minimum):***
> >
> > |  | AES | PS | HPSv2 | IR |
> > | --- | --- | --- | --- | --- |
> > | Base Model A | 3.4223 | 14.5546 | 20.9475 | -1.9042 |
> > | Base Model B | 3.3457 | 14.4538 | 21.1954 | -2.0415 |
> > | Base Model C | 3.4184 | 14.5845 | 21.0943 | -1.9423 |
> > | AFA (Ours) | 3.4583 | 14.4358 | 21.0084 | -1.9647 |
> >
> > ***Quantitative Comparison of ensembling FLUX .1 [dev] models (Maximum):***
> >
> > |  | AES | PS | HPSv2 | IR |
> > | --- | --- | --- | --- | --- |
> > | Base Model A | 8.8740 | 30.4235 | 36.3341 | 2.1394 |
> > | Base Model B | 8.8442 | 30.4623 | 35.6012 | 2.4731 |
> > | AFA (Ours) | **8.8801** | **30.4800** | **36.3984** | 2.4802 |
> >
> > ***Quantitative Comparison of ensembling FLUX .1 [dev] models (Minimum):***
> >
> > |  | AES | PS | HPSv2 | IR |
> > | --- | --- | --- | --- | --- |
> > | Base Model A | 3.3418 | 15.4933 | 22.9432 | -1.5422 |
> > | Base Model B | 3.5941 | 15.3942 | 21.4947 | -1.2473 |
> > | AFA (Ours) | 3.4815 | 15.3974 | 20.3934 | -1.3425 |
> > ---
> > **Q2:** What are the AES score and HPSv2 score of the used JourneyDB dataset?
> >
> > **A2:** The metrics of our training dataset (a 10,000-sample subset of JourneyDB) are presented below. We provide the average, maximum, and minimum values for reference.
> >
> > ***Evaluation Metrics on the training subset of JourneyDB:***
> >
> > |  | AES | PS | HPSv2 | IR |
> > | --- | --- | --- | --- | --- |
> > | Average | 6.3685 | 21.3524 | 27.8115 | 0.8793 |
> > | Maximum | 7.7709 | 26.5971 | 34.0841 | 2.0173 |
> > | Minimum | 3.3764 | 15.9359 | 21.0134 | -2.2622 |

---

### Official Review · Reviewer_RkPx · 2024-11-01

**Soundness:** 4
**Presentation:** 3
**Contribution:** 3
**Rating:** 8
**Confidence:** 3

**Summary:**

The paper introduces Adaptive Feature Aggregation (AFA), a new feature layer ensemble method for generating high-quality images from text. In particular, the paper proposes a Spatial-Aware Block-Wise (SABW) aggregator, which seeks to aggregate intermediate features of various pre-trained models dynamically at the block level. Furthermore, the paper includes a wide range of quantitative & qualitative validations showing strong empirical evidence of AFA's improvements in a range of image generation tasks.

**Strengths:**

The strengths of the paper are as follows:
- The paper is well-written and provides a clear outline of the underlying motivation for AFA.
- The paper provides a timely analysis of an underexplored field of feature layer ensemble methods for diffusion models.
- SABW component addresses a novel spatial attention variation in denoising capabilities across states.
- The resulting AFA shows strong empirical results across a wide range of tasks.

**Weaknesses:**

Some weaknesses of the paper are as follows:
- From the reviewer's perspective, the AFA method, especially the SABW component, introduces a large degree of additional complexity.
- The review would also like to see some empirical comparisons with other methods when evaluating the computational efficiency of AFA.

**Questions:**

Where there any cases or prompts where the authors find that AFA underperforms expectations?

---

> ### Author Response · Authors · 2024-11-23
> **Response to Reviewer RkPx**
>
> Thanks for your valuable comments. We will address the concerns below.
>
> ## Weaknesses
>
> **W1:** AFA, especially SABW, introduces a large degree of additional complexity.
>
> **R1:** SABW does not significantly increase computational complexity. While SABW introduces additional parameters, the increase is minimal—only about 50M parameters, which is just 1/16 of the base model’s parameter count of approximately 800M. Furthermore, due to the high tolerance for fewer inference steps, AFA can ensemble multiple base models and achieve superior generation performance with fewer steps compared to a single base model. In other words, the computational complexity of AFA  remains comparable to that of a single base model when generating images.

---

> ### Author Response · Authors · 2024-11-23
> **Response to Reviewer RkPx**
>
> **W2:** The comparison of computational efficiency is missing.
>
> **R2:** Based on your comment, we compare the efficiency between our AFA and the baselines, when considering 2, 3, and 6 base models.
>
> ***Efficiency Comparison on 2 base models:***
>
> |  | # params. | TFLOPs | Times (s) |
> | --- | --- | --- | --- |
> | Merging Methods | 859.52M | 70.24 | 2.94 |
> | Magic Fusion | 1719.04M | 138.66 | 6.01 |
> | AFA (50 inf. steps) | 1761.47M | 148.64 | 6.28 |
> | AFA (30 inf. steps) | 1761.47M | 94.24 | 3.21 |
>
> ***Efficiency Comparison on 3 base models:***
>
> |  | # params. | TFLOPs | Times (s) |
> | --- | --- | --- | --- |
> | Merging Methods | 859.52M | 70.24 | 2.94 |
> | Magic Fusion | 2578.56M | 206.88 | 9.13 |
> | AFA (50 inf. steps) | 2621.03M | 218.74 | 8.98 |
> | AFA (20 inf. steps) | 2621.03M | 86.61 | 3.62 |
>
> ***Efficiency Comparison on 6 base models:***
>
> |  | # params. | TFLOPs | Times (s) |
> | --- | --- | --- | --- |
> | Merging Methods | 859.52M | 70.24 | 2.94 |
> | Magic Fusion | 5157.12M | 411.53 | 22.41 |
> | AFA (50 inf. steps) | 5210.37M | 430.08 | 23.15 |
> | AFA (10 inf. steps) | 5210.37M | 103.68 | 4.58 |
>
> As shown in the above tables, when using the same 50 inference steps, ensembling methods (e.g., MagicFusion and AFA) do not have an efficiency advantage over a single base model or merging methods. The number of model parameters, TFLOPs, and inference time all increase linearly with the number of base models.
>
> However, thanks to AFA’s high tolerance for fewer inference steps, it can achieve similar performance with reduced steps. Therefore, under fewer inference steps, AFA achieves efficiency in terms of TFLOPs and inference time comparable to that of a single base models and merging methods.
>
> ***IR metrics of AFA with varying the number of base models and inference steps:***
>
> |  | 2 base models | 3 base models | 6 base models |
> | --- | --- | --- | --- |
> | 50 inference steps | 0.5003 | 0.4892 | 0.5347 |
> | 30 inference steps | 0.4791 | — | — |
> | 20 inference steps | — | 0.4803 | — |
> | 10 inference steps | — | — | 0.5135 |
>
> The details can be found in Appendix F and H of the revised manuscript.

---

> ### Author Response · Authors · 2024-11-23
> **Response to Reviewer RkPx**
>
> ## Questions
>
> **Q1:** Did AFA underperform in any specific cases or with certain prompts?
>
> **A1:** Yes. Specifically, when all the base diffusion models produce poor results for a given prompt, AFA’s performance becomes limited. Since AFA depends on aggregating features from these base models, if they all struggle with a particularly challenging prompt, the aggregated output is unlikely to show significant improvement.

---

### Official Review · Reviewer_DhvC · 2024-11-03

**Soundness:** 3
**Presentation:** 2
**Contribution:** 3
**Rating:** 5
**Confidence:** 4

**Summary:**

This study focuses on dynamically ensembling multiple diffusion models with the same architecture for stronger general generation capability, previously done mostly by static merging. The authors propose to train a plug-in module to generate spatial attention maps for each U-Net block to aggregate the intermediate features. Experiments demonstrate consistent performance improvement.

**Strengths:**

1.	The studied task is very useful in practice. The technical contribution of this study should be acknowledged.
2.	The proposed method is intuitive and reasonable, with inner workings and results that match the motivation.
3.	Experiments show consistent performance improvement.

**Weaknesses:**

1.	The presentation really needs polishing. For example, “various states” are repeatedly mentioned in both line 80 and line 85. There is a missing reference in line 352. Some of the fonts in Figure 2 are also too small. The following two weaknesses are also related to presentation.
2.	SABW implements spatial attention using the composition of several modules, instead of a variant of typical multi-head self-attention in diffusion models. This makes the presentation quite confusing for readers who are familiar with diffusion model architecture. I think the authors should clearly remind readers of this difference, as well as explain why such a special architecture is necessary in comparison to some variant of self-attention.
3.	While this study aims to enhance the general generation capability, it is also a common goal of ensembling/merging to combine the styles/concepts of multiple models, which is not the interest of this study. The authors should help readers notice this difference.

**Questions:**

1.	More recent diffusion models, such as Stable Diffusion XL and DiT-based models, have different architectures from Stable Diffusion v1.5. For example, if we view DiT models in the same way as used in this study, the entire model will be a single block. SDXL models also have significantly larger inner structures within each block. How may the proposed method generalize to these new models?
2.	As far as I know, some diffusion models also fine-tune the text encoder alongside U-Net. How may the proposed method apply to such scenarios?
3.	In line 346, it is stated that all the methods generate 4 images. This means 4 images per what? Only 4 images per dataset will cause serious insufficiency in evaluation, so the authors should clarify if this is another presentation issue.

---

> ### Author Response · Authors · 2024-11-23
> **Response to Reviewer DhvC**
>
> Thanks for your valuable comments. We will address the concerns below.
>
> ## Weaknesses
>
> **W1:** Some of the presentations need polishing.
>
> **R1:** Thank you for your suggestion. We have refined the text (line 083-085), corrected the mistakes (line 354), and made Figure 2 easier to read in the revised manuscript.
>
> **W2:** The authors should highlight the difference between the spatial attention of SABW and the typical multi-head self-attention. The authors should explain why the special architecture of SABW is necessary in comparison to some variant of self-attention.
>
> **R2:** Thanks for your insightful suggestions. We have highlighted the difference and explained the necessity in the revised manuscript (line 250-258).
>
> Specifically, the typical multi-head self-attention mechanism simply uses three linear layers to project the input features into three separate spaces(i.e., query, key, and value), and then computes the attention map across all the projected features. Our spatial-attention mechanism, based on an important experimental insight from Figure 1, takes into account the divergent denoising capabilities of multiple diffusion models, which are influenced by the prompts and denoising steps, and change dynamically across spatial locations. To account for this, our projection method incorporates the input prompt and the current denoising step. Moreover, it is sufficient to compute the attention map independently for each spatial location. As a result, better generation performance can be achieved by enabling multiple models to collaborate at different spatial locations under the conditions of the prompt and the denoising step.
>
> **W3:** This study focuses on improving general generation capability, not on combining the styles or concepts of multiple models through ensembling/merging. The authors should highlight this distinction for the readers.
>
> **R3:** Thank you for your insightful suggestion. We have emphasized this point in the revised manuscript (line 139 and 161, i.e., the footnote).

---

> ### Author Response · Authors · 2024-11-23
> **Response to Reviewer DhvC**
>
> ## Questions
>
> **Q1:** How AFA generalize to the models with new diffusion frameworks, such as Stable Diffusion XL and DiT-based models?
>
> **A1:** Based on your comment, we evaluate the generality of our AFA on other diffusion architectures. We select three SDXL [1] models and two FLUX .1 [dev] [2] models from CivitAI.
>
> SDXL consists two U-Nets: the denoising U-Net and the refining U-Net. Both U-Nets shares a similar architecture of SDv1.5, but contain larger blocks. Consistent with the method used for ensembling SDv1.5 models, we employ the SABW module to aggregate features from each block.
>
> ***Quantitative Comparison of ensembling SDXL models:***
>
> |  | AES | PS | HPSv2 | IR |
> | --- | --- | --- | --- | --- |
> | Base Model A | 7.3204 | 28.3445  | 35.5682 | .6314 |
> | Base Model B | 7.7341 | 28.3834 | 35.5793 | .6328 |
> | Base Model C | 7.3475 | 28.4009 | 35.5384 | .6339 |
> | AFA (Ours) | **7.8029** | **28.4513** | **35.6003** | **.6342** |
>
> FLUX .1 [dev] is a DiT-based model. To ensemble models with this architecture, we also utilize the SABW module to aggregate features from each DiT Transformer block.
>
> ***Quantitative Comparison of ensembling FLUX .1 [dev] models:***
>
> |  | AES | PS | HPSv2 | IR |
> | --- | --- | --- | --- | --- |
> | Base Model A | 8.8740 | 30.4235 | 36.3341 | .7341 |
> | Base Model B | 8.8442 | 30.4623 | 35.6012 | .7324 |
> | AFA (Ours) | **8.8801** | **30.4800** | **36.3984** | **.7370** |
>
> The performance of the ensembled model surpasses that of the individual base models, demonstrating the generality of our AFA on different model architectures. However, the observed improvement is relatively small, likely because the base models already exhibit strong performance, and ensembling leads to only marginal gains. Additionally, due to the large size of the base models, ensembling multiple such models may be less practical.
>
> It is worth noting that our AFA cannot be applied to ensembling models with different architectures, as the misalignment of block features makes block-wise aggregation challenging.
> We view ensembling models with diverse architectures as a promising direction for future research.
>
> The details can be found in Appendix I of the revised manuscript.

---

> ### Author Response · Authors · 2024-11-23
> **Response to Reviewer DhvC**
>
> **Q2:** How does AFA handle the diffusion models with different fine-tuned text encoders?
>
> **A2:** Each ensembled diffusion model uses its own fine-tuned text encoder to encode the prompt. The resulting text embeddings are then injected into its U-Net blocks via the cross-attention mechanism. Since AFA aggregates the output features from the U-Net blocks, the fine-tuned text encoders remain unaffected by the feature aggregation process.
>
> **Q3:** What is meant by "generating 4 images"?
>
> **A3:** "Generating 4 images" refers to the process where, for each prompt, the models generate 4 images from different initialized noises.
>
> ---
>
> [1] SDXL: Improving Latent Diffusion Models for High-Resolution Image Synthesis
>
> [2] https://blackforestlabs.ai/announcing-black-forest-labs/

---

> ### Author Response · Authors · 2024-11-26
> **Addressing Incorrect Results for SDXL and FLUX**
>
> We appreciate Reviewer Ynbz for pointing out the issue of weird results for SDXL. Upon re-examination, we discovered that the reported AES, PS, and HPSv2 scores for the SDXL and FLUX architectures were incorrectly calculated due to a coding error (`torch.max()` was used instead of `torch.mean()`). As a result, the values represent the maximum scores rather than the average. We are very sorry for this oversight.
>
> The revised average results for SDXL and FLUX are as follows. Thankfully, the conclusions remain unchanged: the performance of the ensembled model surpasses that of the individual base models. However, the observed improvement is relatively modest.
>
> We have updated our manuscript accordingly in Appendix I.
>
> ***Quantitative Comparison of ensembling SDXL models (Average):***
>
> |  | AES | PS | HPSv2 | IR |
> | --- | --- | --- | --- | --- |
> | Base Model A | 6.1948 | 21.7854 | 28.0452 | 0.6314 |
> | Base Model B | 6.1363 | 21.7783 | 28.0531 | 0.6328 |
> | Base Model C | 6.1852 | 21.7752 | 27.9984 | 0.6339 |
> | AFA (Ours) | **6.2190** | **21.7959** | **28.0894** | **0.6342** |
>
> ***Quantitative Comparison of ensembling FLUX .1 [dev] models (Average):***
>
> |  | AES | PS | HPSv2 | IR |
> | --- | --- | --- | --- | --- |
> | Base Model A | 6.2341 | 21.8843 | 28.0958 | 0.7341 |
> | Base Model B | 6.2346 | 21.8593 | 28.1003 | 0.7324 |
> | AFA (Ours) | **6.2490** | **21.8931** | **28.1194** | **0.7370** |
>
> Additionally, we now also include the maximum and minimum values for completeness, as follows.
>
> ***Quantitative Comparison of ensembling SDXL models (Maximum):***
>
> |  | AES | PS | HPSv2 | IR |
> | --- | --- | --- | --- | --- |
> | Base Model A | 7.3204 | 28.3445  | 35.5682 | 1.9344 |
> | Base Model B | 7.7341 | 28.3834 | 35.5793 | 1.9452 |
> | Base Model C | 7.3475 | 28.4009 | 35.5384 | 1.9573 |
> | AFA (Ours) | **7.8029** | **28.4513** | **35.6003** | **2.0034** |
>
> ***Quantitative Comparison of ensembling SDXL models (Minimum):***
>
> |  | AES | PS | HPSv2 | IR |
> | --- | --- | --- | --- | --- |
> | Base Model A | 3.4223 | 14.5546 | 20.9475 | -1.9042 |
> | Base Model B | 3.3457 | 14.4538 | 21.1954 | -2.0415 |
> | Base Model C | 3.4184 | 14.5845 | 21.0943 | -1.9423 |
> | AFA (Ours) | 3.4583 | 14.4358 | 21.0084 | -1.9647 |
>
> ***Quantitative Comparison of ensembling FLUX .1 [dev] models (Maximum):***
>
> |  | AES | PS | HPSv2 | IR |
> | --- | --- | --- | --- | --- |
> | Base Model A | 8.8740 | 30.4235 | 36.3341 | 2.1394 |
> | Base Model B | 8.8442 | 30.4623 | 35.6012 | 2.4731 |
> | AFA (Ours) | **8.8801** | **30.4800** | **36.3984** | **2.4802** |
>
> ***Quantitative Comparison of ensembling FLUX .1 [dev] models (Minimum):***
>
> |  | AES | PS | HPSv2 | IR |
> | --- | --- | --- | --- | --- |
> | Base Model A | 3.3418 | 15.4933 | 22.9432 | -1.5422 |
> | Base Model B | 3.5941 | 15.3942 | 21.4947 | -1.2473 |
> | AFA (Ours) | 3.4815 | 15.3974 | 20.3934 | -1.3425 |

---

> ### Author Response · Authors · 2024-11-30
>
> Dear Reviewer DhvC,
>
> We have carefully prepared a response to address your concerns. Could you kindly take a moment to review it and let us know if it resolves the issues you raised? If you have any additional questions or suggestions, we would be happy to address them.
>
> Thank you for your time and consideration.
>
> The Authors

---

### Official Review · Reviewer_TNJb · 2024-11-04

**Soundness:** 2
**Presentation:** 3
**Contribution:** 2
**Rating:** 6
**Confidence:** 3

**Summary:**

The paper investigates an ensemble of multiple diffusion models with identical architectures. Rather than simply blending the network parameters of multiple diffusion models through weighted averaging, it introduces a trainable "glue" module, referred to as the AFA method, to integrate them into a cohesive system. This module dynamically adjusts each model's contribution at the feature level, adapting to various conditions such as prompts, initial noise, denoising steps, and spatial locations. Specifically, the paper proposes a U-Net-specific "glue" module, called SABW. Through extensive experiments and ablation studies, the effectiveness of SABW in integrating three diffusion models with U-Net architectures is thoroughly demonstrated.

**Strengths:**

1. The paper is well-written and well-structured, with clear and informative figures and tables that effectively support the content.
2. The motivation is interesting and novel: using a trainable "glue" module to integrate multiple pre-trained diffusion model experts at the feature level, aiming to improve generation quality. This approach is instructive and easy to follow.
3. The experimental design is comprehensive and thorough. The paper conducts extensive experiments on two datasets, COCO 2017 and Draw Bench Prompts, utilizing four evaluation metrics for each. These experiments provide robust evidence of SABW's effectiveness in integrating three diffusion models with U-Net architectures, demonstrating the model's enhanced performance across diverse scenarios.

**Weaknesses:**

1. This paper aims to effectively integrate multiple diffusion models to achieve improved generation quality. However, based on the framework and experimental setup of SABW, it appears that SABW is only suitable for integrating diffusion models with identical U-Net architectures, which may limit its flexibility and applicability.
2. According to the training algorithm presented, SABW can theoretically be used to combine any N diffusion models. However, the experiments seem to only consider the integration of three diffusion models, without sufficient discussion or explanation of this choice. A study on the impact of the number of integrated diffusion models would better validate the effectiveness of SABW.
3. During inference, AFA essentially combines all base models with the glue module SABW, which results in relatively low inference efficiency. The authors provided results for AFA with a reduced number of inference steps compared to the base models. However, comparing AFA's results to those of the baselines under the same reduced-step inference would better validate AFA's practical time efficiency.

**Questions:**

1. Could the authors provide further explanation as to why you chose to randomly split the six diffusion models into two groups of three, rather than integrating all six models directly? Additionally, could you provide more insights into the rationale behind selecting three diffusion models for integration?
2. During the training process of AFA, it appears that intermediate inference features of all base diffusion models at different timesteps are required. Wouldn’t this process be quite time-consuming? Could the authors provide more information on the training time efficiency of AFA compared to other baselines?

---

> ### Author Response · Authors · 2024-11-23
> **Response to Reviewer TNJb**
>
> Thanks for your valuable comments. We will address the concerns below.
>
> ## Weaknesses
>
> **W1:** It appears that SABW is only suitable for integrating diffusion models with identical U-Net architectures, which may limit its flexibility and applicability.
>
> **R1:** Based on your comment, we evaluate the generality of our AFA on other diffusion architectures. We select three SDXL [1] models and two FLUX .1 [dev] [2] models from CivitAI.
>
> SDXL consists two U-Nets: the denoising U-Net and the refining U-Net. Both U-Nets shares a similar architecture of SDv1.5, but contain larger blocks. Consistent with the method used for ensembling SDv1.5 models, we employ the SABW module to aggregate features from each block.
>
> ***Quantitative Comparison of ensembling SDXL models:***
>
> |  | AES | PS | HPSv2 | IR |
> | --- | --- | --- | --- | --- |
> | Base Model A | 7.3204 | 28.3445  | 35.5682 | .6314 |
> | Base Model B | 7.7341 | 28.3834 | 35.5793 | .6328 |
> | Base Model C | 7.3475 | 28.4009 | 35.5384 | .6339 |
> | AFA (Ours) | **7.8029** | **28.4513** | **35.6003** | **.6342** |
>
> FLUX .1 [dev] is a DiT-based model. To ensemble models with this architecture, we also utilize the SABW module to aggregate features from each DiT Transformer block.
>
> ***Quantitative Comparison of ensembling FLUX .1 [dev] models:***
>
> |  | AES | PS | HPSv2 | IR |
> | --- | --- | --- | --- | --- |
> | Base Model A | 8.8740 | 30.4235 | 36.3341 | .7341 |
> | Base Model B | 8.8442 | 30.4623 | 35.6012 | .7324 |
> | AFA (Ours) | **8.8801** | **30.4800** | **36.3984** | **.7370** |
>
> The performance of the ensembled model surpasses that of the individual base models, demonstrating the generality of our AFA on different model architectures. However, the observed improvement is relatively small, likely because the base models already exhibit strong performance, and ensembling leads to only marginal gains. Additionally, due to the large size of the base models, ensembling multiple such models may be less practical.
>
> It is worth noting that our AFA cannot be applied to ensembling models with different architectures, as the misalignment of block features makes block-wise aggregation challenging.
> We view ensembling models with diverse architectures as a promising direction for future research.
>
> The details can be found in Appendix I of the revised manuscript.

---

> > ### Author Response · Authors · 2024-11-26
> > **Addressing Incorrect Results for SDXL and FLUX**
> >
> > We appreciate Reviewer Ynbz for pointing out the issue of weird results for SDXL. Upon re-examination, we discovered that the reported AES, PS, and HPSv2 scores for the SDXL and FLUX architectures were incorrectly calculated due to a coding error (`torch.max()` was used instead of `torch.mean()`). As a result, the values represent the maximum scores rather than the average. We are very sorry for this oversight.
> >
> > The revised average results for SDXL and FLUX are as follows. Thankfully, the conclusions remain unchanged: the performance of the ensembled model surpasses that of the individual base models. However, the observed improvement is relatively modest.
> >
> > We have updated our manuscript accordingly in Appendix I.
> >
> > ***Quantitative Comparison of ensembling SDXL models (Average):***
> >
> > |  | AES | PS | HPSv2 | IR |
> > | --- | --- | --- | --- | --- |
> > | Base Model A | 6.1948 | 21.7854 | 28.0452 | 0.6314 |
> > | Base Model B | 6.1363 | 21.7783 | 28.0531 | 0.6328 |
> > | Base Model C | 6.1852 | 21.7752 | 27.9984 | 0.6339 |
> > | AFA (Ours) | **6.2190** | **21.7959** | **28.0894** | **0.6342** |
> >
> > ***Quantitative Comparison of ensembling FLUX .1 [dev] models (Average):***
> >
> > |  | AES | PS | HPSv2 | IR |
> > | --- | --- | --- | --- | --- |
> > | Base Model A | 6.2341 | 21.8843 | 28.0958 | 0.7341 |
> > | Base Model B | 6.2346 | 21.8593 | 28.1003 | 0.7324 |
> > | AFA (Ours) | **6.2490** | **21.8931** | **28.1194** | **0.7370** |
> >
> > Additionally, we now also include the maximum and minimum values for completeness, as follows.
> >
> > ***Quantitative Comparison of ensembling SDXL models (Maximum):***
> >
> > |  | AES | PS | HPSv2 | IR |
> > | --- | --- | --- | --- | --- |
> > | Base Model A | 7.3204 | 28.3445  | 35.5682 | 1.9344 |
> > | Base Model B | 7.7341 | 28.3834 | 35.5793 | 1.9452 |
> > | Base Model C | 7.3475 | 28.4009 | 35.5384 | 1.9573 |
> > | AFA (Ours) | **7.8029** | **28.4513** | **35.6003** | **2.0034** |
> >
> > ***Quantitative Comparison of ensembling SDXL models (Minimum):***
> >
> > |  | AES | PS | HPSv2 | IR |
> > | --- | --- | --- | --- | --- |
> > | Base Model A | 3.4223 | 14.5546 | 20.9475 | -1.9042 |
> > | Base Model B | 3.3457 | 14.4538 | 21.1954 | -2.0415 |
> > | Base Model C | 3.4184 | 14.5845 | 21.0943 | -1.9423 |
> > | AFA (Ours) | 3.4583 | 14.4358 | 21.0084 | -1.9647 |
> >
> > ***Quantitative Comparison of ensembling FLUX .1 [dev] models (Maximum):***
> >
> > |  | AES | PS | HPSv2 | IR |
> > | --- | --- | --- | --- | --- |
> > | Base Model A | 8.8740 | 30.4235 | 36.3341 | 2.1394 |
> > | Base Model B | 8.8442 | 30.4623 | 35.6012 | 2.4731 |
> > | AFA (Ours) | **8.8801** | **30.4800** | **36.3984** | **2.4802** |
> >
> > ***Quantitative Comparison of ensembling FLUX .1 [dev] models (Minimum):***
> >
> > |  | AES | PS | HPSv2 | IR |
> > | --- | --- | --- | --- | --- |
> > | Base Model A | 3.3418 | 15.4933 | 22.9432 | -1.5422 |
> > | Base Model B | 3.5941 | 15.3942 | 21.4947 | -1.2473 |
> > | AFA (Ours) | 3.4815 | 15.3974 | 20.3934 | -1.3425 |

---

> ### Author Response · Authors · 2024-11-23
> **Response to Reviewer TNJb**
>
> **W2:** A study on the impact of the number of integrated diffusion models would better validate the effectiveness of SABW.
>
> **R2:** Based on your suggestion, we apply AFA to ensemble a larger number of base models.
>
> |  | 2 base models | 3 base models | 4 base models | 5 base models | 6 base models |
> | --- | --- | --- | --- | --- | --- |
> | 50 inf. steps | 0.5003 | 0.4892 | 0.4967 | 0.5042 | 0.5347 |
> | 40 inf. steps | 0.4834 | 0.4874 | 0.4906 | 0.4984 | 0.5310 |
> | 30 inf. steps | 0.4791 | 0.4846 | 0.4853 | 0.4993 | 0.5294 |
> | 20 inf. steps | 0.3641 | 0.4803 | 0.4840 | 0.4975 | 0.5304 |
> | 10 inf. steps | 0.1844 | 0.4013 | 0.4285 | 0.4423 | 0.5135 |
>
> Based on the results, ensembling more base models leads more performance improvements. Additionally, increasing the number of the ensembled base models enhances the tolerance for fewer inference steps. For example, when ensembling 4 or 5 base models, only 20 inference steps are sufficient, and with 6 base models, just 10 inference steps are sufficient. Further details can be found in Appendix F of the revised manuscript.

---

> ### Author Response · Authors · 2024-11-23
> **Response to Reviewer TNJb**
>
> **W3:** Comparing AFA's results to those of the baselines under the same reduced-step inference would better validate AFA's practical time efficiency.
>
> **R3:** Based on your suggestion, we compare our AFA with the baseline methods under fewer inference steps.
>
> ***IR metrics comparison with varying inference steps:***
>
> | inf. steps | 50 | 40 | 30 | 20 | 10 |
> | --- | --- | --- | --- | --- | --- |
> | Wtd. Merging | 0.4148  | 0.3614  | 0.2697  | 0.0256  | -0.6321  |
> | MBW | 0.4159  | 0.3568  | 0.2619  | 0.0155  | -0.6628  |
> | autoMBW | 0.3593  | 0.3352  | 0.2506  | -0.2507  | -0.6944  |
> | MagicFusion | 0.3756  | 0.3620  | 0.3256  | 0.2150  | 0.0497  |
> | AFA | **0.4640**  | **0.4632**  | **0.4542**  | **0.4497**  | **0.3794**  |
>
> As demonstrated by the results, our AFA not only surpasses the baseline methods at higher inference steps but also achieves significantly better performance with fewer inference steps. This highlights AFA's superior tolerance to the fewer inference steps compared to the baselines, resulting in improved inference efficiency. More details can be seen in Appendix G of the revised manuscript.

---

> ### Author Response · Authors · 2024-11-23
> **Response to Reviewer TNJb**
>
> ## Questions
>
> **Q1:** Could the authors provide further explanation as to why you chose to randomly split the six diffusion models into two groups of three, rather than integrating all six models directly?
>
> **A1:** To evaluate the effectiveness of our AFA across different model groups, we randomly selected two groups, each containing three models. The choice of three models for ensembling was arbitrary, and our AFA is fully capable of handling more than three models. To assess the scalability of our AFA, we applied it to ensemble a larger number of base models.
>
> ***IR metrics of AFA with varying the number of base models and inference steps:***
>
> |  | 2 base models | 3 base models | 4 base models | 5 base models | 6 base models |
> | --- | --- | --- | --- | --- | --- |
> | 50 inf. steps | 0.5003 | 0.4892 | 0.4967 | 0.5042 | 0.5347 |
> | 40 inf. steps | 0.4834 | 0.4874 | 0.4906 | 0.4984 | 0.5310 |
> | 30 inf. steps | 0.4791 | 0.4846 | 0.4853 | 0.4993 | 0.5294 |
> | 20 inf. steps | 0.3641 | 0.4803 | 0.4840 | 0.4975 | 0.5304 |
> | 10 inf. steps | 0.1844 | 0.4013 | 0.4285 | 0.4423 | 0.5135 |
>
> As shown in the results above, ensembling more base models leads more performance improvements. Additionally, increasing the number of the ensembled base models enhances the tolerance for fewer inference steps.
>
> More details can be found in Appendix F of the revised manuscript.

---

> ### Author Response · Authors · 2024-11-23
> **Response to Reviewer TNJb**
>
> **Q2 (1):** During the training process of AFA, it appears that intermediate inference features of all base diffusion models at different timesteps are required. Wouldn’t this process be quite time-consuming?
>
> **A2 (1):** During training, only the parameters of the SABW modules are updated, while the parameters of the base models remain frozen. In each training step, a single timestep is sampled, and AFA learns to aggregate features specific to this timestep. Overall, our AFA has a time-efficient training process.
>
> **Q2 (2):** Could the authors provide more information on the training time efficiency of AFA compared to other baselines?
>
> **A2 (2):** The baselines, including the merging-based and MagicFusion, are the training-free methods, making a direct comparison of training efficiency with our AFA infeasible. Specifically, when ensembling three base models to denoise a 512×512 image, the training FLOPs of our AFA amount to 6.42T.
>
> ---
>
> [1] SDXL: Improving Latent Diffusion Models for High-Resolution Image Synthesis
>
> [2] https://blackforestlabs.ai/announcing-black-forest-labs/

---

> ### Author Response · Authors · 2024-11-30
>
> Dear Reviewer TNJb,
>
> We have carefully prepared a response to address your concerns. Could you kindly take a moment to review it and let us know if it resolves the issues you raised? If you have any additional questions or suggestions, we would be happy to address them.
>
> Thank you for your time and consideration.
>
> The Authors

---

> > ### Comment · Reviewer_TNJb · 2024-12-01
> >
> > Thanks to the authors for providing comprehensive and thorough experimental results. My concerns have been addressed, and I will raise my score.

---

> > > ### Author Response · Authors · 2024-12-01
> > >
> > > We sincerely appreciate your recognition and are especially grateful for the increased overall rating.
> > > Thank you for taking the time to review our responses and manuscript revisions, as well as for your insightful feedback that has greatly helped us improve our work.

---

### Official Review · Reviewer_x8ta · 2024-11-04

**Soundness:** 3
**Presentation:** 2
**Contribution:** 2
**Rating:** 6
**Confidence:** 3

**Summary:**

This work introduces a novel method called Adaptive Feature Aggregation (AFA) to enhance the performance of text-guided diffusion models by dynamically adjusting the contributions of multiple models at the feature level. Different from the existing static model which utilizes merging strategies, AFA can recognize the performance of diffusion models depending on different states including text prompts, initial noises, denoising steps, and spatial locations. The method employs a lightweight Spatial-Aware Block-Wise (SABW) feature aggregator, which produces attention maps to realize adaptively aggregation of block-level intermediate features from multiple U-Net denoisers.

**Strengths:**

1. the method can adjust the contributions of multiple diffusion models based on various states.
2. the generated attention maps can give a straightforward view of the context and timestamps in the ensembled diffusion model.
3. AFA demonstrates its tolerance to reductions in inference steps.

**Weaknesses:**

1. the method integrates intermediate features from U-Net denoisers with the same architecture, posing doubts on its ability to ensemble a wider range of model types or architectures.
2. As the number of base models increases, the method's reliance on multiple base models may result in overfitting if, say, the base models have correlated features.

**Questions:**

1. How does the method handle the situation that base models have correlated features or that base model bias is generated by imbalanced data?
2. How is the method’s performance when facing particularly noisy input data?
3. What modifications should be considered when applying AFA to integrate models with different architectures?
4. Are there any potential optimization strategies to reduce the extra computational complexity brought by AFA to every single inference?
5. What if the contributions of different models have conflict impacts on the result significantly?

---

> ### Author Response · Authors · 2024-11-23
> **Response to Reviewer x8ta**
>
> Thanks for your valuable comments. We will address the concerns below.
>
> ## Weaknesses
>
> **W1:** There are concerns about AFA’s capability to ensemble a broader range of model types or architectures.
>
> **R1:** Based on your comment, we evaluate the generality of our AFA on other diffusion architectures. We select three SDXL [1] models and two FLUX .1 [dev] [2] models from CivitAI.
>
> SDXL consists two U-Nets: the denoising U-Net and the refining U-Net. Both U-Nets shares a similar architecture of SDv1.5, but contain larger blocks. Consistent with the method used for ensembling SDv1.5 models, we employ the SABW module to aggregate features from each block.
>
> ***Quantitative Comparison of ensembling SDXL models:***
>
> |  | AES | PS | HPSv2 | IR |
> | --- | --- | --- | --- | --- |
> | Base Model A | 7.3204 | 28.3445  | 35.5682 | .6314 |
> | Base Model B | 7.7341 | 28.3834 | 35.5793 | .6328 |
> | Base Model C | 7.3475 | 28.4009 | 35.5384 | .6339 |
> | AFA (Ours) | **7.8029** | **28.4513** | **35.6003** | **.6342** |
>
> FLUX .1 [dev] is a DiT-based model. To ensemble models with this architecture, we also utilize the SABW module to aggregate features from each DiT Transformer block.
>
> ***Quantitative Comparison of ensembling FLUX .1 [dev] models:***
>
> |  | AES | PS | HPSv2 | IR |
> | --- | --- | --- | --- | --- |
> | Base Model A | 8.8740 | 30.4235 | 36.3341 | .7341 |
> | Base Model B | 8.8442 | 30.4623 | 35.6012 | .7324 |
> | AFA (Ours) | **8.8801** | **30.4800** | **36.3984** | **.7370** |
>
> The performance of the ensembled model surpasses that of the individual base models, demonstrating the generality of our AFA on different model architectures. However, the observed improvement is relatively small, likely because the base models already exhibit strong performance, and ensembling leads to only marginal gains. Additionally, due to the large size of the base models, ensembling multiple such models may be less practical.
>
> The details can be found in Appendix I of the revised manuscript.

---

> > ### Author Response · Authors · 2024-11-26
> > **Addressing Incorrect Results for SDXL and FLUX**
> >
> > We appreciate Reviewer Ynbz for pointing out the issue of weird results for SDXL. Upon re-examination, we discovered that the reported AES, PS, and HPSv2 scores for the SDXL and FLUX architectures were incorrectly calculated due to a coding error (`torch.max()` was used instead of `torch.mean()`). As a result, the values represent the maximum scores rather than the average. We are very sorry for this oversight.
> >
> > The revised average results for SDXL and FLUX are as follows. Thankfully, the conclusions remain unchanged: the performance of the ensembled model surpasses that of the individual base models. However, the observed improvement is relatively modest.
> >
> > We have updated our manuscript accordingly in Appendix I.
> >
> > ***Quantitative Comparison of ensembling SDXL models (Average):***
> >
> > |  | AES | PS | HPSv2 | IR |
> > | --- | --- | --- | --- | --- |
> > | Base Model A | 6.1948 | 21.7854 | 28.0452 | 0.6314 |
> > | Base Model B | 6.1363 | 21.7783 | 28.0531 | 0.6328 |
> > | Base Model C | 6.1852 | 21.7752 | 27.9984 | 0.6339 |
> > | AFA (Ours) | **6.2190** | **21.7959** | **28.0894** | **0.6342** |
> >
> > ***Quantitative Comparison of ensembling FLUX .1 [dev] models (Average):***
> >
> > |  | AES | PS | HPSv2 | IR |
> > | --- | --- | --- | --- | --- |
> > | Base Model A | 6.2341 | 21.8843 | 28.0958 | 0.7341 |
> > | Base Model B | 6.2346 | 21.8593 | 28.1003 | 0.7324 |
> > | AFA (Ours) | **6.2490** | **21.8931** | **28.1194** | **0.7370** |
> >
> > Additionally, we now also include the maximum and minimum values for completeness, as follows.
> >
> > ***Quantitative Comparison of ensembling SDXL models (Maximum):***
> >
> > |  | AES | PS | HPSv2 | IR |
> > | --- | --- | --- | --- | --- |
> > | Base Model A | 7.3204 | 28.3445  | 35.5682 | 1.9344 |
> > | Base Model B | 7.7341 | 28.3834 | 35.5793 | 1.9452 |
> > | Base Model C | 7.3475 | 28.4009 | 35.5384 | 1.9573 |
> > | AFA (Ours) | **7.8029** | **28.4513** | **35.6003** | **2.0034** |
> >
> > ***Quantitative Comparison of ensembling SDXL models (Minimum):***
> >
> > |  | AES | PS | HPSv2 | IR |
> > | --- | --- | --- | --- | --- |
> > | Base Model A | 3.4223 | 14.5546 | 20.9475 | -1.9042 |
> > | Base Model B | 3.3457 | 14.4538 | 21.1954 | -2.0415 |
> > | Base Model C | 3.4184 | 14.5845 | 21.0943 | -1.9423 |
> > | AFA (Ours) | 3.4583 | 14.4358 | 21.0084 | -1.9647 |
> >
> > ***Quantitative Comparison of ensembling FLUX .1 [dev] models (Maximum):***
> >
> > |  | AES | PS | HPSv2 | IR |
> > | --- | --- | --- | --- | --- |
> > | Base Model A | 8.8740 | 30.4235 | 36.3341 | 2.1394 |
> > | Base Model B | 8.8442 | 30.4623 | 35.6012 | 2.4731 |
> > | AFA (Ours) | **8.8801** | **30.4800** | **36.3984** | **2.4802** |
> >
> > ***Quantitative Comparison of ensembling FLUX .1 [dev] models (Minimum):***
> >
> > |  | AES | PS | HPSv2 | IR |
> > | --- | --- | --- | --- | --- |
> > | Base Model A | 3.3418 | 15.4933 | 22.9432 | -1.5422 |
> > | Base Model B | 3.5941 | 15.3942 | 21.4947 | -1.2473 |
> > | AFA (Ours) | 3.4815 | 15.3974 | 20.3934 | -1.3425 |

---

> ### Author Response · Authors · 2024-11-23
> **Response to Reviewer x8ta**
>
> **W2:** As the number of base models increases, the method's reliance on multiple base models may result in overfitting if, say, the base models have correlated features.
>
> **R2:** Based on your comment, we design an experiment to evaluate the robustness of our AFA against highly correlated features.
>
> Specifically, we selected one high-quality base model and one low-quality base model. Using AFA, we ensemble the high-quality model with several copied low-quality models to evaluate whether the ensembled model will perform toward that of the low-quality model due to the dominance of the correlated features from the low-quality model.
>
> |  | AES | PS | HPSv2 | IR |
> | --- | --- | --- | --- | --- |
> | High-Quality Base Model (H) | 5.5641  | 21.8013 | 28.0183 | .4238 |
> | Low-Quality Base Model (L) | 5.5013  | 21.4624  | 27.7246 | .2835 |
> | AFA (1H + 1L) | 5.6013  | 21.9341 | 28.3485 | .4746 |
> | AFA (1H + 2L) | 5.6000 | 21.9348 | 28.3398 | .4739 |
> | AFA (1H + 3L) | 5.5974 | 21.8248 | 28.3399 | .4730 |
> | AFA (1H + 5L) | 5.5831 | 21.8019  | 28.2035 | .4593 |
>
> Based on the results, although the performance the ensembled model slightly declines when ensembling with large number of low-quality models, it still outperforms the high-quality model.
> This demonstrates that our AFA exhibits strong robustness to highly correlated features.
>
> Details can be found in Appendix L of the revised manuscript.

---

> ### Author Response · Authors · 2024-11-23
> **Response to Reviewer x8ta**
>
> ## Questions
>
> **Q1:** How does the method handle the situation that base models have correlated features or that base model bias is generated by imbalanced data?
>
> **A1:** The training goal of our AFA is to achieve better denoising capability by learning a spatial attention map to linearly aggregate the features from the base models. During inference, given a specific prompt, the SABW module adaptively assigns weights to aggregate the features of the base models, striving to achieve best denoising capability.
>
> Regardless of whether the base models exhibit correlated features or biases, the SABW module assigns higher weights to features that contribute positively to the denoising process and lower weights to those that may hinder the denoising process.
>
> **Q2:** How is the method’s performance when facing particularly noisy input data?
>
> **A2:** According to the principles of diffusion models, the image generation process begins by sampling a fully noisy latent representation that follows the standard Gaussian distribution. Consequently, during inference, our method only needs to handle noise conforming to this distribution. While different initial noise samples may introduce some diversity, the generated images will remain robust and faithfully adhere to the context of the given prompt.
>
> **Q3:** What modifications should be considered when applying AFA to integrate models with different architectures?
>
> A3: Due to the misalignment of block features, our AFA cannot be directly applied to ensemble models with different architectures. Achieving this would likely require an additional feature alignment method. For instance, inspired by REPA [3], a contrastive learning objective could be employed to align features from models with different architectures. We believe this direction holds significant potential and warrants further research.
>
> **Q4:** Are there any potential optimization strategies to reduce the extra computational complexity brought by AFA to every single inference?
>
> **A4:** In Appendix E of the original manuscript, we explore another MoE-based method, which applies the SABW module to determine the next block to be executed among the base models. While this method has the potential to further reduce the computational complexity of each inference step, we identified several disadvantages:
>
> - ***Poor Performance.*** As shown in Table 4 and 5 of Appendix E, the performance of the MoE methods is  inferior to that of our AFA. Additionally, the MoE methods do not support fewer inference steps.
> - ***Comparable Training Computation Efficiency.*** Since all experts are still required to participate in training, the training TFLOPs of the MoE methods are comparable to that of our AFA.
> - ***Inability to Perform Parallel Inference.*** Since the classifier-free guidance (CFG, Section 3.5) activates two different routes, the parallel inference cannot be conducted, which will increase the inference time. Furthermore, when multiple images need to be generated simultaneously, the MoE-based methods are inefficient, because the activated route for each image must be processed individually.
>
> Exploring other alternative ensembling strategies to reduce the computational complexity of each inference step is a promising direction for further research.
>
> **Q5:** What if the contributions of different models have conflict impacts on the result significantly?
>
> **A5:** Given a prompt, if the base models have significantly conflict impacts on the result, it indicates that some base models contribute positively for the denoising process, while some have a negative impact. In such cases, the well-trained SABW module assigns higher weights to the features of the positive base models and lower weights to those of the negative base models.
>
> ---
>
> [1] SDXL: Improving Latent Diffusion Models for High-Resolution Image Synthesis
>
> [2] https://blackforestlabs.ai/announcing-black-forest-labs/
>
> [3] Representation Alignment for Generation: Training Diffusion Transformers Is Easier Than You Think

---

> ### Author Response · Authors · 2024-11-30
>
> Dear Reviewer x8ta,
>
> We have carefully prepared a response to address your concerns. Could you kindly take a moment to review it and let us know if it resolves the issues you raised? If you have any additional questions or suggestions, we would be happy to address them.
>
> Thank you for your time and consideration.
>
> The Authors

---

> > ### Comment · Reviewer_x8ta · 2024-12-03
> >
> > Thanks for clarifying my concern, I am satisfied and will increase my score accordingly.

---

> > > ### Author Response · Authors · 2024-12-03
> > >
> > > We sincerely appreciate your recognition and are especially grateful for the increased overall rating. Thank you for taking the time to review our responses and manuscript revisions, as well as for your insightful feedback that has greatly helped us improve our work.

---

### Official Review · Reviewer_WgNL · 2024-11-06

**Soundness:** 2
**Presentation:** 3
**Contribution:** 3
**Rating:** 5
**Confidence:** 3

**Summary:**

This paper introduces a novel ensembling method called Adaptive Feature Aggregation (AFA) for leveraging multiple high-quality diffusion models to enhance text-to-image generation capabilities. The key innovation of AFA is its ability to dynamically adjust the contributions of multiple models at the feature level based on various states, such as prompts, initial noises, denoising steps, and spatial locations. This is achieved through a lightweight Spatial-Aware Block-Wise (SABW) feature aggregator that adaptively aggregates block-wise intermediate features from multiple U-Net denoisers into a unified one.

The paper's main contributions are:
1. The proposal of an ensembling-based AFA method that dynamically adjusts the contributions of multiple models at the feature level.
2. The design of the SABW feature aggregator, which can produce attention maps according to various states to adaptively aggregate the block-level intermediate features from multiple U-Net denoisers.
3. The demonstration, through both quantitative and qualitative experiments, that the proposed AFA outperforms the base models and the baseline methods in terms of superior image quality and context alignment.

**Strengths:**

The paper presents a novel Adaptive Feature Aggregation (AFA) method for ensembling multiple diffusion models in text-guided image generation.

1. The paper introduces a new approach to ensembling diffusion models by dynamically adjusting the contributions of multiple models at the feature level based on various states such as prompts, initial noises, denoising steps, and spatial locations. This is a departure from existing methods that primarily adopt parameter merging strategies to produce a new static model.

2. The proposed AFA method is evaluated quantitatively and qualitatively, demonstrating improvements in image quality and context alignment. The method outperforms base models and baseline methods.

3. The paper is well-structured, with clear explanations of the proposed method, including the Spatial-Aware Block-Wise (SABW) feature aggregator. The authors provide a detailed description of the AFA framework, its components, and the training and inference processes.

4. The paper addresses an important research direction in leveraging multiple high-quality diffusion models to improve text-to-image generation. The proposed AFA method has the potential to significantly enhance the generation capabilities of diffusion models, making it a valuable contribution to the field.

**Weaknesses:**

1. AFA's computational efficiency is lower for individual inference steps compared to merging-based methods, primarily due to the additional parameters introduced by the Spatial-Aware Block-Wise (SABW) feature aggregator. However, the paper notes that AFA's tolerance for fewer inference steps can offset this initial inefficiency, making the overall computational cost comparable to that of base models or merging methods.

2. The performance of AFA is contingent on the quality of the base models. If the base models have inherent limitations in their generative capabilities, increasing the number of training samples for AFA does not necessarily lead to improved performance.

3. While the paper proposes a dynamic feature-level ensembling method, it does not extensively explore alternative ensemble strategies that could potentially offer further improvements.

4. The paper does not address the scalability of the AFA method when ensembling a larger number of models, which could become computationally prohibitive.

**Questions:**

Has any research optimizing the SABW module to reduce its computational overhead been done? potentially through model pruning, quantization, or more efficient network architectures.

Other ensemble strategies can be compared, such as model distillation or more sophisticated attention mechanisms.

Could the authors extend their evaluations to diverse datasets to testify its generality?

---

> ### Author Response · Authors · 2024-11-23
> **Response to Reviewer WgNL**
>
> Thanks for your valuable comments. We will address the concerns below.
>
> ## Weaknesses
>
> **W1:** AFA's efficiency is lower for individual inference steps compared to merging-based methods. However, AFA is tolerance for fewer inference steps can offset this initial inefficiency.
>
> **R1:** Compared to merging-based methods, our AFA does not offer advantages in a single inference step. However, since image generation requires multiple inference steps, a fair comparison should evaluate efficiency of the entire generation process.
>
> Based on the experimental results, our AFA achieves comparable performance with fewer inference steps.
>
> ***IR metrics of AFA with varying the number of base models and inference steps:***
>
> |  | 2 base models | 3 base models | 6 base models |
> | --- | --- | --- | --- |
> | 50 inference steps | 0.5003 | 0.4892 | 0.5347 |
> | 30 inference steps | 0.4791 | — | — |
> | 20 inference steps | — | 0.4803 | — |
> | 10 inference steps | — | — | 0.5135 |
>
> Building on this, we compare the efficiency of the merging-based methods and our AFA using 2, 3, and 6 base models.
>
> ***Efficiency Comparison on 2 base models:***
>
> |  | # params. | TFLOPs | Times (s) |
> | --- | --- | --- | --- |
> | Base Models | 859.52M | 70.24 | 2.94 |
> | AFA (50 inf. steps) | 1761.47M | 148.64 | 6.28 |
> | AFA (30 inf. steps) | 1761.47M | 94.24 | 3.21 |
>
> ***Efficiency Comparison on 3 base models:***
>
> |  | # params. | TFLOPs | Times (s) |
> | --- | --- | --- | --- |
> | Base Models | 859.52M | 70.24 | 2.94 |
> | AFA (50 inf. steps) | 2621.03M | 218.74 | 8.98 |
> | AFA (20 inf. steps) | 2621.03M | 86.61 | 3.62 |
>
> ***Efficiency Comparison on 6 base models:***
>
> |  | # params. | TFLOPs | Times (s) |
> | --- | --- | --- | --- |
> | Base Models | 859.52M | 70.24 | 2.94 |
> | AFA (50 inf. steps) | 5210.37M | 430.08 | 23.15 |
> | AFA (10 inf. steps) | 5210.37M | 103.68 | 4.58 |
>
> Under fewer inference steps, AFA demonstrate efficiency in terms of TFLOPs and inference time comparable to that of a single base model.
>
> The details can be found in Appendix F and H of the revised manuscript.

---

> ### Author Response · Authors · 2024-11-23
> **Response to Reviewer WgNL**
>
> **W2:** The performance of AFA is contingent on the quality of the base models.
>
> **R2:** The performance of the ensembled model being limited by the quality of the base models is a broad and inherent limitation in model ensembling. A straightforward way to mitigate this issue is to select high-quality base models, which, fortunately, is not cost-prohibitive.
>
> **W3:** This paper does not extensively explore alternative ensemble strategies that could potentially offer further improvements.
>
> **R3:** In Appendix E of the original manuscript, we explore another MoE-based method, which applies the SABW module to determine the next block to be executed among the base models. While this method has the potential to further improve performance and reduce FLOPs, we identified several disadvantages:
>
> - ***Poor Performance.*** As shown in Table 4 and 5 of Appendix E, the performance of the MoE methods is  inferior to that of our AFA. Additionally, the MoE methods do not support fewer inference steps.
> - ***Comparable Training Computation Efficiency.*** Since all experts are still required to participate in training, the training TFLOPs of the MoE methods are comparable to that of our AFA.
> - ***Inability to Perform Parallel Inference.*** Since the classifier-free guidance (CFG, Section 3.5) activates two different routes, the parallel inference cannot be conducted, which will increase the inference time. Furthermore, when multiple images need to be generated simultaneously, the MoE-based methods are inefficient, because the activated route for each image must be processed individually.
>
> Exploring other alternative ensembling strategies is a promising direction for further research.

---

> ### Author Response · Authors · 2024-11-23
> **Response to Reviewer WgNL**
>
> **W4:** The paper does not address the scalability of the AFA method when ensembling a larger number of models, which could become computationally prohibitive.
>
> **R4:** Based on your comment, we apply AFA to ensemble a larger number of base models.
>
> |  | 2 base models | 3 base models | 4 base models | 5 base models | 6 base models |
> | --- | --- | --- | --- | --- | --- |
> | 50 inf. steps | 0.5003 | 0.4892 | 0.4967 | 0.5042 | 0.5347 |
> | 40 inf. steps | 0.4834 | 0.4874 | 0.4906 | 0.4984 | 0.5310 |
> | 30 inf. steps | 0.4791 | 0.4846 | 0.4853 | 0.4993 | 0.5294 |
> | 20 inf. steps | 0.3641 | 0.4803 | 0.4840 | 0.4975 | 0.5304 |
> | 10 inf. steps | 0.1844 | 0.4013 | 0.4285 | 0.4423 | 0.5135 |
>
> Based on the results, ensembling more base models leads more performance improvements. Additionally, increasing the number of the ensembled base models enhances the tolerance for fewer inference steps. For example, when ensembling 4 or 5 base models, only 20 inference steps are sufficient, and with 6 base models, just 10 inference steps are sufficient. Further details can be found in Appendix F of the revised manuscript.

---

> ### Author Response · Authors · 2024-11-23
> **Response to Reviewer WgNL**
>
> ## Questions
>
> **Q1:** Has any research optimizing the SABW module to reduce its computational overhead been done? potentially through model pruning, quantization, or more efficient network architectures.
>
> **A1:** In Appendix E the original manuscript, we explore the MoE-based method that uses the SABW module to determine the next block to be executed among the base models. This method can be viewed as a form of model pruning. While this method has the potential to reduce computational overhead, it suffers from poor performance.
>
> Since the computational overhead of SABW is lower than that of the U-Net denoisers, applying quantization and designing more efficient architectures for SABW could not significantly improve the computational overhead of AFA. Nonetheless, we acknowledge that optimizing of the efficiency of AFA remains necessary and is a worthwhile direction for further research.

---

> ### Author Response · Authors · 2024-11-23
> **Response to Reviewer WgNL**
>
> **Q2:** Other ensemble strategies can be compared, such as model distillation or more sophisticated attention mechanisms.
>
> **A2:** Based on your suggestion, we attempted to further distill the ensembled model into significantly fewer steps. Specifically, we explore using LCM [1] to achieve this. In our experiments, we kept the parameters of the denoisers frozen while training only the parameters of the SABW modules. However, this approach is unsuccessful. A possible reason for this failure is that the frozen parameters may have impeded the distillation process. The details can be found in Appendix J of the revised manuscript.
>
> ***Experimental results for distilling into fewer inference steps using LCM:***
>
> | Inference Steps | 20 | 4 | 2 | 1 |
> | --- | --- | --- | --- | --- |
> | IR | 0.4793 | *-0.5846* | *-0.5594* | *-0.5952* |

---

> ### Author Response · Authors · 2024-11-23
> **Response to Reviewer WgNL**
>
> **Q3:** Could the authors extend their evaluations to diverse datasets to testify its generality?
>
> **A3:** Based on your question, we evaluate AFA on three additional datasets: DiffusionDB [2], JourneyDB [3], and LAION-COCO [4]. The results are as follows. Please note that we present the averaged results of Group I and II. The experimental settings, the full experimental results, and comparisons with the baselines can be found in Appendix K of the revised manuscript. These results demonstrate that AFA consistently outperforms both the base models, highlighting the robustness and effectiveness of AFA.
>
> ***Quantitative comparison on DiffusionDB:***
>
> |  | **FID** | **IS** | **CLIP-I** | **CLIP-T** | **AES** | **PS** | **HPSv2** | **IR** |
> | --- | --- | --- | --- | --- | --- | --- | --- | --- |
> | **Base Model A** | 14.55 | 6.95 | .6506 | .3056 | 5.585 | 23.4820 | 30.0324 | .4384 |
> | **Base Model B** | 15.00 | 6.97 | .6566 | .3069 | 5.479 | 23.2126 | 29.9083 | .4265 |
> | **Base Model C** | 15.08 | 6.91 | .6416 | .2986 | 5.480 | 23.3422 | 29.9377 | .4315 |
> | **AFA (Ours)** | **13.15** | **7.25** | **.6739** | **.3451** | **5.736** | **23.5631** | **30.1439** | **.4738** |
>
> ***Quantitative comparison on JourneyDB:***
>
> |  | **FID** | **IS** | **CLIP-I** | **CLIP-T** | **AES** | **PS** | **HPSv2** | **IR** |
> | --- | --- | --- | --- | --- | --- | --- | --- | --- |
> | **Base Model A** | 17.50 | 5.69 | .5351 | .2498 | 4.549 | 19.1577 | 24.5596 | .3619 |
> | **Base Model B** | 18.09 | 5.72 | .5403 | .2544 | 4.474 | 18.9517 | 24.4556 | .3470 |
> | **Base Model C** | 17.93 | 5.61 | .5220 | .2416 | 4.476 | 19.0699 | 24.4344 | .3499 |
> | **AFA (Ours)** | **15.79** | **5.91** | **.5534** | **.2854** | **4.707** | **19.2655** | **24.6398** | **.3920** |
>
> ***Quantitative comparison on LAION-COCO:***
>
> |  | **FID** | **IS** | **CLIP-I** | **CLIP-T** | **AES** | **PS** | **HPSv2** | **IR** |
> | --- | --- | --- | --- | --- | --- | --- | --- | --- |
> | **Base Model A** | 13.54 | 7.94 | .7438 | .3495 | 6.331 | 26.6869 | 34.1564 | .05021 |
> | **Base Model B** | 14.00 | 7.97 | .7490 | .3542 | 6.225 | 26.3973 | 34.0227 | .4832 |
> | **Base Model C** | 13.96 | 7.82 | .7282 | .3375 | 6.225 | 26.5178 | 34.0177 | .4883 |
> | **AFA (Ours)** | **12.25** | **8.26** | **.7717** | **.3988** | **6.567** | **26.8175** | **34.2971** | **.5368**  |
>
> ---
>
> [1] Latent Consistency Models: Synthesizing High-Resolution Images with Few-Step Inference
>
> [2] DiffusionDB: A Large-Scale Prompt Gallery Dataset for Text-to-Image Generative Models
>
> [3] JourneyDB: A Benchmark for Generative Image Understanding
>
> [4] https://laion.ai/blog/laion-coco/

---

> ### Author Response · Authors · 2024-11-30
>
> Dear Reviewer WgNL,
>
> We have carefully prepared a response to address your concerns. Could you kindly take a moment to review it and let us know if it resolves the issues you raised? If you have any additional questions or suggestions, we would be happy to address them.
>
> Thank you for your time and consideration.
>
> The Authors

---

> > ### Comment · Reviewer_WgNL · 2024-12-03
> >
> > Thanks to the authors for addressing my concern, and I will raise my score to 6.

---

> > > ### Author Response · Authors · 2024-12-03
> > >
> > > Thank you for taking the time to review our responses and manuscript revisions, as well as for providing insightful feedback that has significantly improved our work.
> > >
> > > We are sincerely grateful for your recognition and especially appreciate your willingness to consider raising the score.
> > >
> > > However, we have noticed that the score in the system has not yet been updated.
> > > If possible, could you kindly update it at your earliest convenience?
> > > Your support in this matter would be greatly appreciated, as it could have a meaningful impact on the paper decision.

---

### Official Review · Reviewer_W7j2 · 2024-11-07

**Soundness:** 3
**Presentation:** 3
**Contribution:** 3
**Rating:** 8
**Confidence:** 4

**Summary:**

This paper introduces Adaptive Feature Aggregation (AFA), a novel ensembling method for text-to-image diffusion models. Unlike conventional static parameter merging approaches, AFA dynamically aggregates features from multiple diffusion models using a lightweight Spatial-Aware Block-Wise (SABW) feature aggregator. The method adaptively adjusts model contributions based on various states including prompts, initial noise, denoising steps, and spatial locations. The key technical innovations lie in block-wise feature aggregation and spatial attention mechanisms, enabling effective ensemble that maximizes strengths while minimizing weaknesses of individual models.

**Strengths:**

- Novel perspective on model ensembling that goes beyond both conventional parameter merging and timestep-specific expert selection strategies
- Comprehensive consideration of multiple states (prompts, noise levels, timesteps, spatial locations) for adaptive feature aggregation, offering more fine-grained control than existing expert-based approaches
- Well-designed empirical validation with thorough ablation studies and clear visualization of spatial attention patterns
- Practical advantages in terms of implementation, requiring only the training of SABW while keeping base models frozen
- Demonstrates robust performance with fewer inference steps, suggesting potential for real-world applications
Strong quantitative improvements across multiple metrics (FID, IS, CLIP scores) compared to both single models and existing ensemble methods

**Weaknesses:**

- While the paper introduces a novel method, it lacks direct comparison with recent mixture-of-expert diffusion approaches such as ERNIE-ViLG 2.0 [a], eDiff-I [b], and MEME [c], which also address the dynamic nature of the denoising process with multiple models. Basically, I think that the reader should be informed about the ideas that can be thought of in the direction of improving DPMs using multiple models, such as using different expert models along the time axis, and how this study proposes a different concept from those studies. Specifically, comparing performance metrics (e.g., FID, IS, CLIP scores), computational efficiency (inference times, resource utilization), and qualitative differences in generated images would provide a clearer understanding of how AFA stands relative to these methods. Including such comparisons or ablation studies would strengthen the claim that spatial-aware aggregation offers advantages over or complements timestep-specific expert selection. In the process of improving DPMs by utilizing multiple models, I believe that research directions along two axes, spatial and temporal, should be compared.
- Missing theoretical analysis on when and why spatial-aware feature aggregation outperforms timestep-specific expert selection. Incorporating theoretical insights—such as analyzing the information flow, gradient propagation, or the capacity of spatial attention mechanisms within the diffusion process—could help elucidate the conditions under which AFA is most effective. A comparative theoretical framework would enhance the understanding of the benefits and limitations of spatial versus temporal adaptation strategies.
- Computational overhead during inference could be problematic for practical applications, yet alternatives or optimizations are not thoroughly explored. I think the limitations associated with these computational costs should be clearly explained and guidelines for practical application should be provided.
- Experiments limited to SDv1.5-based models, raising questions about generalizability across different model architectures
- Lack of analysis on the trade-off between the granularity of adaptation (spatial/temporal) and computational cost

---
[a] Feng, Zhida et al. “ERNIE-ViLG 2.0: Improving Text-to-Image Diffusion Model with Knowledge-Enhanced Mixture-of-Denoising-Experts.” 2023 IEEE/CVF Conference on Computer Vision and Pattern Recognition (CVPR) (2022): 10135-10145.

[b] Balaji, Yogesh et al. “eDiff-I: Text-to-Image Diffusion Models with an Ensemble of Expert Denoisers.” ArXiv abs/2211.01324 (2022): n. pag.

[c] Lee, Y., J. Kim, H. Go, M. Jeong, S. Oh, and S. Choi. “Multi-Architecture Multi-Expert Diffusion Models”. Proceedings of the AAAI Conference on Artificial Intelligence, vol. 38, no. 12, Mar. 2024, pp. 13427-36, doi:10.1609/aaai.v38i12.29245.

**Questions:**

- Given that recent work in diffusion models increasingly focuses on efficiency, have you explored knowledge distillation or other compression techniques to reduce the inference overhead of AFA?
- How does the spatial attention mechanism in SABW interact with different types of prompts? Are there certain prompt patterns where timestep-specific expert selection might be more appropriate than spatial aggregation?

---

> ### Author Response · Authors · 2024-11-23
> **Response to Reviewer W7j2**
>
> Thanks for your valuable comments. We will address the concerns below.
>
> ## Weaknesses
>
> **W1:** It lacks comparison with the MoE diffusion approaches.
>
> **R1:** Since the denoising experts in ERNIE-ViLG 2.0 [1], eDiff-I [2], and MEME [3] are specifically trained for particular timestep intervals, they are not compatible with the model ensembling setting, where the denoisers from the base models remain frozen. As an alternative, in Appendix E of the original manuscript, we briefly compared AFA with an intuitive Block-Level MoE method. To follow the methods of the papers that you mentioned, we additionally compare with the Denoiser-Level MoE method in the Appendix E of the revised manuscript. Specifically, we apply our SABW module to select the experts using the Gumbel softmax and the reparameterization trick. Since only one base model is activated during inferencing, the MoE methods may be more efficiency for inference.
>
> However, we have identified several drawbacks of the MoE-based methods:
>
> - ***Poor Performance.*** As shown in Table 4 and 5 of Appendix E, the performance of the MoE methods is  inferior to that of our AFA. Additionally, the MoE methods do not support fewer inference steps.
> - ***Comparable Training Computation Efficiency.*** Since all experts are still required to participate in training, the training TFLOPs of the MoE methods are comparable to that of our AFA.
> - ***Inability to Perform Parallel Inference.*** Since the classifier-free guidance (CFG, Section 3.5) activates two different routes, the parallel inference cannot be conducted, which will increase the inference time. Furthermore, when multiple images need to be generated simultaneously, the MoE-based methods are inefficient, because the activated route for each image must be processed individually.
>
> ***Quantitative comparison with 50 inference steps (showing the average results of Groups I and II):***
>
> |  | AES | PS | **HPSv2** | **IR** |
> | --- | --- | --- | --- | --- |
> | **Base Model A** | 5.4872 | 21.7146 | 27.9095 | .3891 |
> | **Base Model B** | 5.5020 | 21.5937 | 27.8795 | .3519 |
> | **Base Model C** | 5.5297 | 21.6484 | 27.8871 | .3645 |
> | **Denoiser-Level MoE** | 5.5183 | 21.6212 | 27.6594 | .4200 |
> | **Block-Level MoE** | 5.5378 | 21.7663 | 27.9296 | .4250 |
> | **AFA (Ours)** | **5.5500** | **21.8161** | **28.0053** | **.4640** |
>
> ***Quantitative comparison with 20 inference steps:***
>
> |  | AES | PS | **HPSv2** | **IR** |
> | --- | --- | --- | --- | --- |
> | **Denoiser-Level MoE** | 1.8433 | 3.3079 | 4.7577 | .0211 |
> | **Block-Level MoE** | 2.1294 | 3.8471 | 5.0418 | .0221 |
> | **AFA (Ours)** | **5.5137** | **21.7962** | **27.8561** | **.4497** |
>
> ***Efficiency comparison:***
>
> |  | #params. | T-TFLOPs | I-TLOPs (1 image) | Times (s) (1 image) | I-TLOPs (2 images) | Times (s) (2 images) | I-TLOPs (4 images) | Times (s) (4 images) |
> | --- | --- | --- | --- | --- | --- | --- | --- | --- |
> | Base Model | 859.52M | — | 70.24 | 2.94 | 140.45 | 5.61 | 280.88 | 6.87 |
> | Denoiser-Level MoE | 2581.01M | 6.36 | 72.37 | 5.36 | 144.75 | 10.37 | 289.48 | 20.81 |
> | **Block-Level MoE** | 2621.03M | 6.43 | 74.41 | 5.39 | 148.82 | 10.43 | 297.65 | 21.00 |
> | **AFA (50 inf. steps)** | 2621.03M | 6.42 | 218.74 | 8.98 | 437.48 | 15.57 | 874.96 | 21.33 |
> | **AFA (20 inf. steps)** | 2621.03M | 6.42 | 86.61 | 3.62 | 173.22 | 6.41 | 346.45 | 8.74 |
>
> Detailed results and analysis are provided in Appendix E of the revised manuscript. Additionally, we have included introductions of the papers you mentioned in Appendix E of the revised manuscript.

---

> ### Author Response · Authors · 2024-11-23
> **Response to Reviewer W7j2**
>
> **W2:** Missing theoretical analysis on when and why spatial-aware feature aggregation outperforms timestep-specific expert selection.
>
> **R2:** In our response for Weakness #1, the MoE methods (i.e., the timestep-specific expert selection method) underperform our AFA (i.e., the spatial-aware feature aggregation method). In fact, the MoE method can be considered a specific case of our AFA. If, at each inference step, the learned weights from our AFA assign a value of 1 to all the features of a base model and 0 to others, this scenario effectively becomes an expert selection along the temporal axis. However, unlike the MoE methods, our AFA can flexibly assign weights to features from all base models, enabling a more refined and adaptive feature aggregation. This flexibility is crucial for achieving superior performance.

---

> ### Author Response · Authors · 2024-11-23
> **Response to Reviewer W7j2**
>
> **W3:** The limitations associated with the computational costs should be clearly explained and guidelines for practical application should be provided.
>
> **R3:** Based on your suggestion, we compare the efficiency of merging methods and ensembling methods when considering 2, 3, and 6 base models.
>
> ***Efficiency Comparison on 2 base models:***
>
> |  | # params. | TFLOPs | Times (s) |
> | --- | --- | --- | --- |
> | Merging Methods | 859.52M | 70.24 | 2.94 |
> | Magic Fusion | 1719.04M | 138.66 | 6.01 |
> | AFA (50 inf. steps) | 1761.47M | 148.64 | 6.28 |
> | AFA (30 inf. steps) | 1761.47M | 94.24 | 3.21 |
>
> ***Efficiency Comparison on 3 base models:***
>
> |  | # params. | TFLOPs | Times (s) |
> | --- | --- | --- | --- |
> | Merging Methods | 859.52M | 70.24 | 2.94 |
> | Magic Fusion | 2578.56M | 206.88 | 9.13 |
> | AFA (50 inf. steps) | 2621.03M | 218.74 | 8.98 |
> | AFA (20 inf. steps) | 2621.03M | 86.61 | 3.62 |
>
> ***Efficiency Comparison on 6 base models:***
>
> |  | # params. | TFLOPs | Times (s) |
> | --- | --- | --- | --- |
> | Merging Methods | 859.52M | 70.24 | 2.94 |
> | Magic Fusion | 5157.12M | 411.53 | 22.41 |
> | AFA (50 inf. steps) | 5210.37M | 430.08 | 23.15 |
> | AFA (10 inf. steps) | 5210.37M | 103.68 | 4.58 |
>
> As shown in the above tables, when using the same 50 inference steps, ensembling methods (e.g., MagicFusion and AFA) do not have an efficiency advantage over a single base model or merging methods. The number of model parameters, TFLOPs, and inference time all increase linearly with the number of base models.
>
> However, thanks to AFA’s high tolerance for fewer inference steps, it can achieve similar performance with reduced steps, while the baseline methods fail to maintain performance.
>
> ***IR metrics of AFA with varying the number of base models and inference steps:***
>
> |  | 2 base models | 3 base models | 6 base models |
> | --- | --- | --- | --- |
> | 50 inference steps | 0.5003 | 0.4892 | 0.5347 |
> | 30 inference steps | 0.4791 | — | — |
> | 20 inference steps | — | 0.4803 | — |
> | 10 inference steps | — | — | 0.5135 |
>
> ***IR metrics comparison with varying inference steps:***
>
> | inf. steps | 50 | 40 | 30 | 20 | 10 |
> | --- | --- | --- | --- | --- | --- |
> | Wtd. Merging | 0.4148  | 0.3614  | 0.2697  | 0.0256  | -0.6321  |
> | MBW | 0.4159  | 0.3568  | 0.2619  | 0.0155  | -0.6628  |
> | autoMBW | 0.3593  | 0.3352  | 0.2506  | -0.2507  | -0.6944  |
> | MagicFusion | 0.3756  | 0.3620  | 0.3256  | 0.2150  | 0.0497  |
> | AFA | **0.4640**  | **0.4632**  | **0.4542**  | **0.4497**  | **0.3794**  |
>
> Therefore, under fewer inference steps, AFA achieves efficiency in terms of TFLOPs and inference time comparable to that of a single base models and merging methods. For practical applications, we recommend using fewer inference steps when ensembling more base models to enhance inference efficiency.
>
> The details can be found in Appendix F, G, and H of the revised manuscript.

---

> ### Author Response · Authors · 2024-11-23
> **Response to Reviewer W7j2**
>
> **W4:** There is uncertainty regarding the generalizability across different model architectures.
>
> **R4:** Based on your comment, we evaluate the generality of our AFA on other diffusion architectures. We select three SDXL [4] models and two FLUX .1 [dev] [5] models from CivitAI.
>
> SDXL consists two U-Nets: the denoising U-Net and the refining U-Net. Both U-Nets shares a similar architecture of SDv1.5, but contain larger blocks. Consistent with the method used for ensembling SDv1.5 models, we employ the SABW module to aggregate features from each block.
>
> ***Quantitative Comparison of ensembling SDXL models:***
>
> |  | AES | PS | HPSv2 | IR |
> | --- | --- | --- | --- | --- |
> | Base Model A | 7.3204 | 28.3445  | 35.5682 | .6314 |
> | Base Model B | 7.7341 | 28.3834 | 35.5793 | .6328 |
> | Base Model C | 7.3475 | 28.4009 | 35.5384 | .6339 |
> | AFA (Ours) | **7.8029** | **28.4513** | **35.6003** | **.6342** |
>
> FLUX .1 [dev] is a DiT-based model. To ensemble models with this architecture, we also utilize the SABW module to aggregate features from each DiT Transformer block.
>
> ***Quantitative Comparison of ensembling FLUX .1 [dev] models:***
>
> |  | AES | PS | HPSv2 | IR |
> | --- | --- | --- | --- | --- |
> | Base Model A | 8.8740 | 30.4235 | 36.3341 | .7341 |
> | Base Model B | 8.8442 | 30.4623 | 35.6012 | .7324 |
> | AFA (Ours) | **8.8801** | **30.4800** | **36.3984** | **.7370** |
>
> The performance of the ensembled model surpasses that of the individual base models, demonstrating the generality of our AFA on different model architectures. However, the observed improvement is relatively small, likely because the base models already exhibit strong performance, and ensembling leads to only marginal gains. Additionally, due to the large size of the base models, ensembling multiple such models may be less practical.
>
> It is worth noting that our AFA cannot be applied to ensembling models with different architectures, as the misalignment of block features makes block-wise aggregation challenging.
> We view ensembling models with diverse architectures as a promising direction for future research.
>
> The details can be found in Appendix I of the revised manuscript.

---

> > ### Author Response · Authors · 2024-11-26
> > **Addressing Incorrect Results for SDXL and FLUX**
> >
> > We appreciate Reviewer Ynbz for pointing out the issue of weird results for SDXL. Upon re-examination, we discovered that the reported AES, PS, and HPSv2 scores for the SDXL and FLUX architectures were incorrectly calculated due to a coding error (`torch.max()` was used instead of `torch.mean()`). As a result, the values represent the maximum scores rather than the average. We are very sorry for this oversight.
> >
> > The revised average results for SDXL and FLUX are as follows. Thankfully, the conclusions remain unchanged: the performance of the ensembled model surpasses that of the individual base models. However, the observed improvement is relatively modest.
> >
> > We have updated our manuscript accordingly in Appendix I.
> >
> > ***Quantitative Comparison of ensembling SDXL models (Average):***
> >
> > |  | AES | PS | HPSv2 | IR |
> > | --- | --- | --- | --- | --- |
> > | Base Model A | 6.1948 | 21.7854 | 28.0452 | 0.6314 |
> > | Base Model B | 6.1363 | 21.7783 | 28.0531 | 0.6328 |
> > | Base Model C | 6.1852 | 21.7752 | 27.9984 | 0.6339 |
> > | AFA (Ours) | **6.2190** | **21.7959** | **28.0894** | **0.6342** |
> >
> > ***Quantitative Comparison of ensembling FLUX .1 [dev] models (Average):***
> >
> > |  | AES | PS | HPSv2 | IR |
> > | --- | --- | --- | --- | --- |
> > | Base Model A | 6.2341 | 21.8843 | 28.0958 | 0.7341 |
> > | Base Model B | 6.2346 | 21.8593 | 28.1003 | 0.7324 |
> > | AFA (Ours) | **6.2490** | **21.8931** | **28.1194** | **0.7370** |
> >
> > Additionally, we now also include the maximum and minimum values for completeness, as follows.
> >
> > ***Quantitative Comparison of ensembling SDXL models (Maximum):***
> >
> > |  | AES | PS | HPSv2 | IR |
> > | --- | --- | --- | --- | --- |
> > | Base Model A | 7.3204 | 28.3445  | 35.5682 | 1.9344 |
> > | Base Model B | 7.7341 | 28.3834 | 35.5793 | 1.9452 |
> > | Base Model C | 7.3475 | 28.4009 | 35.5384 | 1.9573 |
> > | AFA (Ours) | **7.8029** | **28.4513** | **35.6003** | **2.0034** |
> >
> > ***Quantitative Comparison of ensembling SDXL models (Minimum):***
> >
> > |  | AES | PS | HPSv2 | IR |
> > | --- | --- | --- | --- | --- |
> > | Base Model A | 3.4223 | 14.5546 | 20.9475 | -1.9042 |
> > | Base Model B | 3.3457 | 14.4538 | 21.1954 | -2.0415 |
> > | Base Model C | 3.4184 | 14.5845 | 21.0943 | -1.9423 |
> > | AFA (Ours) | 3.4583 | 14.4358 | 21.0084 | -1.9647 |
> >
> > ***Quantitative Comparison of ensembling FLUX .1 [dev] models (Maximum):***
> >
> > |  | AES | PS | HPSv2 | IR |
> > | --- | --- | --- | --- | --- |
> > | Base Model A | 8.8740 | 30.4235 | 36.3341 | 2.1394 |
> > | Base Model B | 8.8442 | 30.4623 | 35.6012 | 2.4731 |
> > | AFA (Ours) | **8.8801** | **30.4800** | **36.3984** | **2.4802** |
> >
> > ***Quantitative Comparison of ensembling FLUX .1 [dev] models (Minimum):***
> >
> > |  | AES | PS | HPSv2 | IR |
> > | --- | --- | --- | --- | --- |
> > | Base Model A | 3.3418 | 15.4933 | 22.9432 | -1.5422 |
> > | Base Model B | 3.5941 | 15.3942 | 21.4947 | -1.2473 |
> > | AFA (Ours) | 3.4815 | 15.3974 | 20.3934 | -1.3425 |

---

> > > ### Comment · Reviewer_W7j2 · 2024-11-27
> > >
> > > Through the authors' thoughtful explanations and quantitative analyses, my concerns have been substantially addressed. I also appreciate how well they have incorporated the feedback from the review process. Given these improvements, I am inclined to increase my score.

---

> > > > ### Author Response · Authors · 2024-11-28
> > > >
> > > > Thank you for taking the time to review our responses and manuscript revisions, as well as for your insightful feedback that has greatly helped us improve our work.
> > > > We sincerely appreciate your recognition and are especially grateful for the increased overall rating.

---

> ### Author Response · Authors · 2024-11-23
> **Response to Reviewer W7j2**
>
> **W5:** Lack of analysis on the trade-off between the granularity of adaptation (spatial/temporal) and computational cost.
>
> **R5:** In our response to Weakness #1, we pointed out that while the MoE methods (i.e., timestep-specific expert-selection methods) outperform our AFA (i.e., spatial-aware feature-aggregation method) in inference TFLOPs, they offer no advantages in terms of model parameters, training TFLOPs, or inference time.

---

> ### Author Response · Authors · 2024-11-23
> **Response to Reviewer W7j2**
>
> ## Questions
>
> **Q1:** Have you explored knowledge distillation or other compression techniques to reduce the inference overhead of AFA?
>
> **A1:** Based on your suggestion, we attempted to further distill the ensembled model into significantly fewer steps. Specifically, we explore using LCM [6] to achieve this. In our experiments, we kept the parameters of the denoisers frozen while training only the parameters of the SABW modules. However, this approach is unsuccessful. A possible reason for this failure is that the frozen parameters may have impeded the distillation process. Exploring additional techniques to reduce the inference overhead of AFA remains a promising direction for future research. The details can be found in Appendix J of the revised manuscript.
>
> ***Experimental results for distilling into fewer inference steps using LCM:***
> | Inference Steps | 20 | 4 | 2 | 1 |
> | --- | --- | --- | --- | --- |
> | IR | 0.4793 | *-0.5846* | *-0.5594* | *-0.5952* |

---

> ### Author Response · Authors · 2024-11-23
> **Response to Reviewer W7j2**
>
> **Q2:** (1) How does the spatial attention mechanism in SABW interact with different types of prompts? (2) Are there certain prompt patterns where timestep-specific expert selection might be more appropriate than spatial aggregation?
>
> A2: (1) All prompts interact with the SABW module through the cross-attention mechanism within its Transformer layer. (2) In our response to Weakness #1, we demonstrate that, under most prompt patterns, our AFA (i.e., spatial-aware feature-aggregation method) statistically outperforms the MoE methods (i.e., timestep-specific expert-selection methods). Identifying the specific prompt patterns where MoE may outperform AFA warrants further investigation.
>
> ---
>
> [1] ERNIE-ViLG 2.0: Improving Text-to-Image Diffusion Model with Knowledge-Enhanced Mixture-of-Denoising-Experts
>
> [2] eDiff-I: Text-to-Image Diffusion Models with an Ensemble of Expert Denoisers
>
> [3] Multi-Architecture Multi-Expert Diffusion Models
>
> [4] SDXL: Improving Latent Diffusion Models for High-Resolution Image Synthesis
>
> [5] https://blackforestlabs.ai/announcing-black-forest-labs/
>
> [6] Latent Consistency Models: Synthesizing High-Resolution Images with Few-Step Inference

---

### Author Response · Authors · 2024-12-04
**General Comment**

We'd like to again thank all the reviewers for their constructive suggestions and for engaging in helpful discussions. Their comments on our work are much appreciated and help the work be stronger. Below, we summarize the revisions we have made in response to the reviews:

- Thanks to Reviewer `W7j2`. We further improve the experimental settings and provide a more comprehensive analysis of the MoE-based ensembling method in **Appendix E** of the revised manuscript.
- Thanks to Reviewers `W7j2`, `WgNL`, `RkPx`, and `Ynbz`. We add a detailed computational efficiency analysis and comparison in **Appendix H** of the revised manuscript.
- Thanks to Reviewers `W7j2`, `x8ta`, `TNjb`, `RkPx`, and `Ynbz`. We include new experiments on ensembling SDXL and FLUX models in **Appendix I** of the revised manuscript.
- Thanks to Reviewers `W7j2` and `WgNL`. We conduct an additional experiment of using LCM to distill the model into fewer inference steps in **Appendix J** of the revised manuscript.
- Thanks to Reviewers `WgNL` and `TNjb`. We expand our experiments to ensemble more base models in **Appendix F** of the revised manuscript.
- Thanks to Reviewer `WgNL`. We evaluate our method on additional datasets in **Appendix K** of the revised manuscript.
- Thanks to Reviewer **x8ta**. We discuss the robustness of our method against highly correlated features in **Appendix L** of the revised manuscript.
- Thanks to Reviewer `TNjb`. We compare the performance of our method with the baselines under same limited number of inference steps in **Appendix G** of the revised manuscript.
- Thanks to Reviewer `Dhvc`. We polish the texts and clarify key conceptual differences in the **main text** of the revised manuscipt.
- Thanks to Reviewer `Ynbz`. We provide a more detailed description of the experimental settings in **Appendix** **N** of the revised manuscript.
- Thanks to Reviewer `Ynbz`. We discuss how the quality of training data impacts our method in **Appendix M** of the revised manuscript.

**We thank the reviewers for their comments and responses. We did our best to address the concerns raised by reviewers, and we appreciate these improvements could be considered. We also have added these results and findings to our revised manuscript.**

---

### Meta-Review · Area_Chair_saXM · 2024-12-17

**Metareview:**

This paper explores adaptive feature aggregation strategies for effectively ensembling diffusion models. It initially receives borderline scores on average. The authors did hard work during the rebuttal phase. Most of the reviewers are now positive toward this paper. It is non-trivial to rebuttal with as many as seven reviewers within a short time. Above all, I'm happy to recommend accepting this paper.

**Additional Comments On Reviewer Discussion:**

`W7j2`: very positive (8), appreciates the authors'  thoughtful explanations and quantitative analyses during the rebuttal phase.

`WgNL`: borderline (5), claimed to raise the score but didn't. I've checked the rebuttal details. I think the detailed experiments presented in the rebuttal can address the major issue of efficiency and scalability.

`x8ta`: who's satisfied with the rebuttal and increased the score to 6

`TNJb`: appreciated the thorough experimental results presented during rebuttal, increased the score to 6

`DhvC`:  borderline (5), didn't take part in the rebuttal. I evaluate the authors' responses myself. I think the major issue is about the writing and generalization ability. I believe the new experiments are sufficient to address the issue.

`RkPx`: positive (8)

`Ynbz`: positive (6) The reviewer didn't reply to the authors' last comments, where they discovered a mistake in the reported metrics. The authors have corrected the data and updated the paper correspondingly.

Most of the issues are addressed. I recommend accepting this paper.

---

### Decision · Program_Chairs · 2025-01-22

Accept (Poster)